# Fluid Inclusion Studies of Barite Disseminated in Hydrothermal Sediments of the Mohns Ridge

**Marina D. Kravchishina** [1,*] **, Vsevolod Yu. Prokofiev** [1,2] **, Olga M. Dara** [1] **, Boris V. Baranov** [1] **,**
**Alexey A. Klyuvitkin** [1] **, Karina S. Iakimova** [1,3] **, Vladislav Yu. Kalgin** [1] **and Alla Yu. Lein** [1]

[1] Shirshov Institute of Oceanology, Russian Academy of Sciences, 36 Nakhimovsky Prosp.,
Moscow 117997, Russia; vpr2004@rambler.ru (V.Y.P.); olgadara@mail.ru (O.M.D.); bbaranov@ocean.ru (B.V.B.);
klyuvitkin@ocean.ru (A.A.K.); yakikarina@gmail.com (K.S.I.); kalgin.vladislav@yandex.ru (V.Y.K.);
allaulein@gmail.com (A.Y.L.)

[2] Institute of Geology of Ore Deposits, Petrography, Mineralogy, and Geochemistry,
Russian Academy of Sciences, 35 Staromonetny Per., Moscow 119017, Russia

[3] Geological Faculty, Lomonosov Moscow State University, GSP-1, Leninskie Gory, Moscow 119991, Russia

* Correspondence: kravchishina@ocean.ru; Tel.: +7-499-1247737

**Abstract:** This article discusses the results of a fluid inclusion studies in barite collected at the Jan Mayen vent field area (Troll Wall and Perle and Bruse) and Loki's Castle vent field on the Mohns Ridge segment of the Arctic Mid-Ocean Ridge. Three mafic-hosted volcanogenic massive sulfide deposits were examined within the active vent fields that adequately correspond to the geological settings of ultraslow-spreading ridges and P–T conditions. Hydrothermal sediments were investigated to determine the temperature and salinity of the fluids responsible for barite precipitation. The hydrothermal origin of the barite was confirmed by its morphology. Fluid inclusions are two-phase and homogenize into the liquid phase on heating at temperatures below 287 °C. The salt concentration in fluids trapped in inclusions is 2.6–4.4 wt.% NaCl eq. The crystallization temperatures varied from 276 °C to 119 °C and from 307 °C to 223 °C for the Jan Mayen and Loki's Castle vent fields, respectively. The data obtained allowed us to confirm evidence of fluid phase separation in the hydrothermal systems and to expand our knowledge of the temperature and salinity of mineral fluids previously known from recent direct measurements during the cruises within the G.O. Sars research vessel. The fluid inclusions data obtained from barites emphasize the fluid features characteristic of volcanogenic massive sulfide deposits, the similarities and differences among the studied hydrothermal sites and allow comparisons with similar products from other active hydrothermal systems.

**Keywords:** barite; fluid inclusion; hydrothermal vent field; hydrothermal fluid; mineralization; volcanogenic massive sulfide deposits; ultraslow-spreading ridge; Mohns Ridge; Arctic Mid-Ocean Ridge



## 1. Introduction

The major dissolved particulate transformation of barium (Ba) in the ocean is related to the mineral barite (BaSO$_4$) of the natural sulfate class, which is characterized by the orthorhombic singony. Barite is one of the most abundant authigenic minerals in deep ocean sediments [1] and one of a few authigenic minerals that may form in the water column during the microbial oxidation of sinking organic matter (e.g., [2,3]). Two main genetic types of barite form in the marine realm: (i) sedimentary barite–particulate crystals formed usually due to passive biogeochemical precipitation in seawater (pelagic barite) and more rarely through an active biological pathway by several planktonic organisms (biogenic barite) [4]; and (ii) diagenetic or fluid-related barite formed via diagenetic processes with the participation of pore water of sediments or reduced fluids of cold seeps and hydrothermal vents [1,5,6]. The diagenetic type, in our view, includes barites formed from the reduced solutions during their mixing with the oxidized ocean water. Such solutions are divided

into pore waters in sediments, cold methane-rich water solutions in cold seeps, and hot hydrothermal solutions in mid-ocean rift valleys.

The size and morphology of barite crystals formed by those different precipitation pathways are distinct [5]. The formation of pelagic barite is closely linked to the formation of particulate organic-rich aggregates in the mesopelagic zone of the ocean and heterotrophic remineralization of organic matter in particle-associated microenvironments [3,7]. The microcrystals of pelagic barite, ranging in size from 200 nm to 2 μm, are ubiquitous in the ocean and are typically ellipsoidal in shape [4,7]. Diagenetic barite may form at the water–sediment interface or below the seafloor from reduced pore water of sediments when mixed with an oxidized seawater-hosted sulfate ion ($SO_4^{2-}$), e.g., [8]. A similar pathway led to the formation of cold seeps and hydrothermal barite; therefore, different Ba-rich fluids are discharged through seeps or ridge systems into sulfate-rich seawater, and barite saturation is exceeded [9,10]. Diagenetic barite crystals are large (20–700 μm), flat, tabular-shaped, and appear as barite beds in the sedimentary column [5]. Hydrothermal barite is formed by the discharge of modified seawater-derived hydrothermal fluids heated by underlying magmas. Recent studies [11–13] indicated that barite in hydrothermal systems did not necessarily precipitate from a mixture of ambient seawater and a hydrothermal fluid, and microbial sulfate reduction is also taking place in hydrothermal environments. Hydrothermal barite crystals are also larger in size (~10–700 μm) and are typically precipitated as cross-cutting tabular crystals commonly forming rosettes (roses, druses, brooms, and others) [5,6]. One can also see barite intergrowths as compact plates, granular aggregates, and zoned nodules. Hydrothermal barite precipitates from mantle-derived fluids influenced by hydrothermal processes. Ba is leached from the oceanic crust (basalt), and when the circulating fluids interact and mix with sulfate-rich seawater, barite may precipitate. The sulfur (S) isotope ratio of hydrothermal barite is usually a mixture of present-day seawater S and hydrothermal origin S ($H_2S$ oxidation), e.g., [5]. It is well known, e.g., [14], that strontium (Sr) is a common trace substitution in barite structure. Hydrothermal barite is characterized by the Sr isotope ($^{87}Sr/^{86}Sr$) ratio, which often occupies an intermediate position between the modern value in seawater and the isotopic composition in pure hydrothermal fluids [15]. Furthermore, the precipitation of barite at the hydrothermal vent fields is confined to the redox barrier and related to a formation mechanism similar to that of the diagenetic pathway [6].

Seawater, as a rule, is a Ba-unsaturated environment [16]. The barite dissolution rates in seawater, as well as those of sphalerite (often paragenetically associated with barite) and anhydrite, depend on the size of the crystals, but they are generally extremely lower than, for example, the solubility of carbonates [1]. Barite behaves as a closed system under typical oxic seafloor conditions and is not prone to diagenetic alteration, e.g., [14]. Hydrothermal barite occurs in the ocean mainly in low- and medium-temperature hydrothermal sediments, chimneys, nodules, and metalliferous sediments [17,18]. Barium concentrations in the primary (end member) hot solutions of hydrothermal vent fields in the Mid-Atlantic Ridge (MAR) and East Pacific Rise (EPR) differ ~20 times depending on the depth and solution temperature, as well as the composition of the initial igneous rocks (felsic rocks, basalts, or ultramafic rocks), and can reach 789 μmol [19]. Some Ba enters the plume in the water column, while ~5% of hydrothermal Ba flux precipitates together with sulfides and nonmetallic minerals in the composition of edifices, sometimes as barite chimneys such as in the Loki's Castle vent field, Mohns Ridge [13]. The recent constraints on hydrothermal Ba compositions [20] enable the hydrothermal input of Ba to Atlantic deep waters to be assessed at 3%–9% of the observed Ba.

Hydrothermal vent fields in the ocean are divided into several varieties depending on their location relative to the rift system (rifts in the mid-ocean ridge, back-arc spreading, and intraplate volcanism zones), as well as on the ocean depth and, hence, on the pressure and temperature (P–T) conditions and composition of the initial igneous, felsic to ultramafic, rocks [19]. In addition to black smokers, gray and white smokers are also identified in the ocean. The temperature of hydrothermal solutions in black smokers varies from 185 to 363 °C. Such temperatures are unfavorable for barite formation. Low and medium

temperatures (~20–200 °C) and ocean depths of about 1500 m are most favorable for hydrothermal barite precipitation [13,18,21–25].

Fluids are trapped as fluid inclusions (FIs) in some minerals formed during the hydrothermal process [26–29]. Barite crystals may represent a combination of two formation processes involving hot fluids and pore water in the sediment covered ridge system that interact and mix with sulfate-rich seawater [5,13,30]. Samples of barite collected in the Guaymas Basin provide such an example where the fluids are expelled into marine sediment strata where sulfate reduction takes place, depleting the pore-water sulfate from the light S isotope and resulting in barite with S isotope ratios slightly greater than seawater [30]. In this regard, when studying FIs in barites, the question should arise each time about the feasibility of using them for the determination of temperature and salinity of pure (mantle-derived) hydrothermal fluids. There are a number of problems that are frequently encountered in the study of FIs, and many of these problems are critical for the validity of the FIs data and their interpretations [26–29,31]. Nevertheless, barite crystals preserve geochemical fingerprints associated with conditions of formation and are especially important as a key structural component of hydrothermal edifices on the seafloor [17,18].

The high preservation potential of barite and its resistance to diagenetic alteration after burial in oxic settings make this authigenic mineral favorable for studies of the origin of formation fluids [1]. Barite has no retrograde solubility, unlike relatively fast-soluble anhydrite, and persists in hydrothermal structures longer than anhydrite. On the Mohns Ridge, several successful attempts have been made to directly measure the temperature of hydrothermal fluids, i.e., [32–34]. However, these measurements are challenging and require confirmation by other methods. Another reliable approach is FI microthermometry in hydrothermal minerals, particularly barites [27,35,36]. Previous studies [37] investigated the FIs in anhydrite within the Fåvne hydrothermal vent field recently discovered at the Mohns Ridge. There are no reports of FIs hosted in barite within the hydrothermal vent fields studied along the Mohns Ridge [38,39]. To address the lack of published information on the hydrothermal FIs, we performed microthermometric measurements of FIs hosted in barite within the mineral assemblages of the active Troll Wall, Perle and Bruse and Loki's Castle vent fields. Our data on FIs are integrated with published direct measurements of hydrothermal fluids on the Mohns Ridge and both direct and FIs measurements on the MAR. The present study discusses data on the morphology and FIs analysis of barites collected from hydrothermally altered (hydrothermal) sediments with the primary aim of improving the understanding of the genetic aspects and local physicochemical conditions for the formation of sulfate–sulfide deposits on the Mohns Ridge, the sector of the northern MAR extending beyond the Arctic Circle latitude (66° N).

## 2. Geological Setting

Depending on the source rocks and the geological setting, the compositions of the sulfide–sulfate deposits can vary considerably [40]. The geological setting where barite forms determines the geochemistry of the precipitated mineral and its usefulness for various applications [1]. The Mohns segment of the ultraslow-spreading Arctic Mid-Ocean Ridge (AMOR) between 71.2° N and 73.5° N with a 16 mm·yr$^{-1}$ full spreading rate [41] is ~600 km long and extends from the Jan Mayen Fracture Zone to the Knipovich Ridge in the Norwegian–Greenland Sea (Figure 1). Characteristic structural features of the Mohns Ridge are axial volcanic ridges (AVRs) located inside its rift valley [42]. AVRs are strong indicators of an underlying heat source whose heat transfer plays a major role in hydrothermal circulations. On the Mohns Ridge, the AVRs are oriented in an azimuth 30–35° suborthogonally to the spreading direction, while the rift valley extends in the direction of 60° [43–45].

Research into the geology, geochemistry, and biota of the various hydrothermal vent fields on the Mohns Ridge began in 2005 [32,46]. A feature of the Mohns Ridge is the presence of active hydrothermal fields at different depths of the rift valley, varying in terms of fluid temperature (from ~20 °C to 320 °C) and mineral composition (from non-

metallic vent fields to polymetallic ore-rich deposits) that are developed on bare mafic substrate. The magma chambers located beneath these chimneys have temperatures of around 1100–1200 °C [13,37].

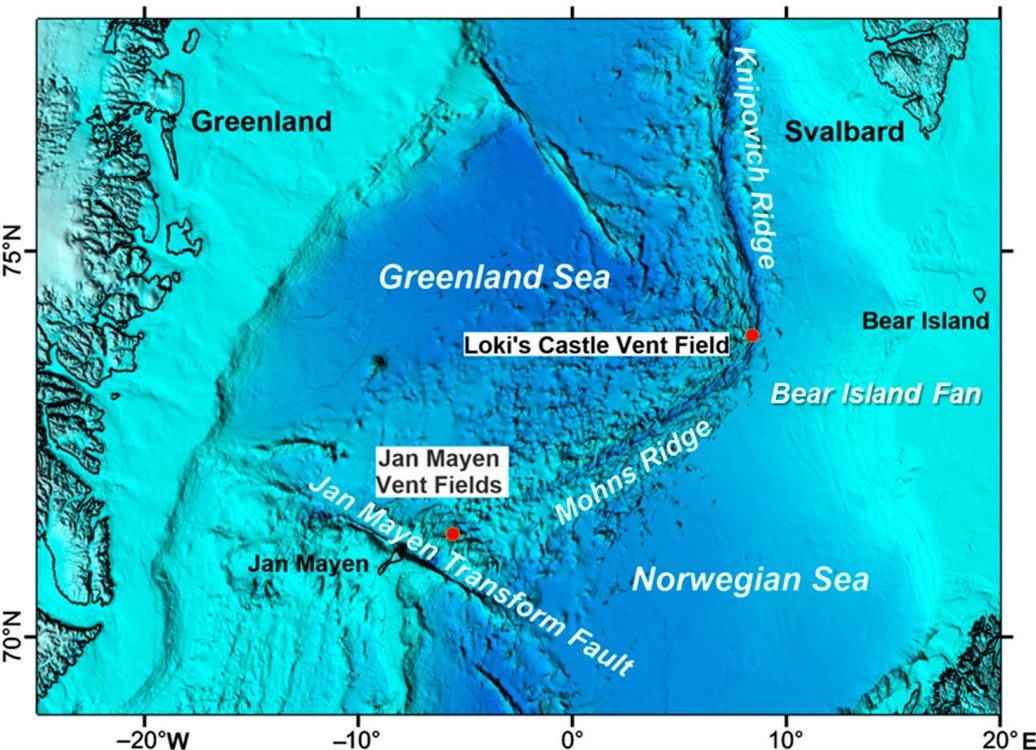

**Figure 1.** General location of the Mohns Ridge, Jan Mayen vent fields, and Loki's Castle vent field (based on GEBCO bathymetry). Red dots indicate the location of the studied sites.

Therefore, several active areas of vents have been discovered and studied in the Mohns Ridge: (i) at upper bathyal (from 550 to 724 m) depths—the Jan Mayen vent field area at 71° N; (ii) at deep bathyal (from 2200 to ~3000 m) depths—the Loki's Castle, AEgir and Fåvne sulfide vent fields at 72–73° N [32–34,37,38,45,47–49]. The active areas of vents are usually located at the floor of the rift valley and are related to the large AVRs. In addition to the active fields, several inactive sites with associated seafloor massive sulfide deposits were discovered along the Mohns Ridge, e.g., Copper Hill at 900 m depth, the Gnitahei deposit at ~3000 m depth, and Mohns Treasure at 2600 m depth at 72–73° N [32,37,50].

The southernmost segment of the Mohns Ridge is affected by the Iceland and Jan Mayen hotspots and is characterized by anomalously high magmatic activity, an over-thickened crust, and shallow water depths [48,51]. Several active hydrothermal vent fields have been discovered in 2005–2014 on the Jan Mayen AVR summit at a relatively shallow depth [32,37], namely: Troll Wall, Soria Moria I and II, Perle and Bruse, and the inactive field Gallionella Garden. All these fields are collectively referred to as the Jan Mayen vent fields, and the shallow southernmost part of the ridge is termed the Jan Mayen vent field area [52].

The northernmost segment of the Mohns Ridge is characterized by less magmatic activity and crustal thickness than the southern segment of the ridge; spreading centers and rift valleys become deeper and more pronounced, and low-angle detachment faults with up to ~10 km displacements are locally developed [52]. The active Loki's Castle vent field was first discovered in 2008 [32]. This black smoker vent field is associated with massive seafloor sulfide deposits. Dome-shaped core complexes that locally expose lower crustal and upper mantle rocks occur on the western flank of the AVR, whereas the eastern flank is covered by sediments belonging to the distal parts of the Bear Island sediment fan [33,37].

## 3. Materials and Methods

### 3.1. Study Area

Bathymetric data of the Mohns Ridge, collected by IO RAS during the 75th cruise of the RV Akademik Mstislav Keldysh in 2019 [53,54], allows us to distinguish 15 AVRs in the Mohns rift valley. Among them, the Jan Mayen AVR is the largest and is located near the Jan Mayen hotspot [51], which is one of the most active areas of submarine neo-volcanism in the ocean.

The sediment samples considered in this work for the FI study were obtained at three stations within the Jan Mayen vent field area (Figure 2a, left inset). Station 6137 was located at the base of the rift valley slope; stations 6131 and 5516 were located on one of the normal rift valley scarps; and all of them are near the Troll Wall vent field (Figure 2b). Station 6146 was subjected to the Perle and Bruse vent field and was located at a distance of about 2 km northeast of the Fe-Mn crust field [53] (Figure 2a, left inset). The samples of station 6147 (Figure 2a, right inset, and Figure 2c) were directly obtained within the Loki's Castle vent field confined to a rift at the AVR summit [33].

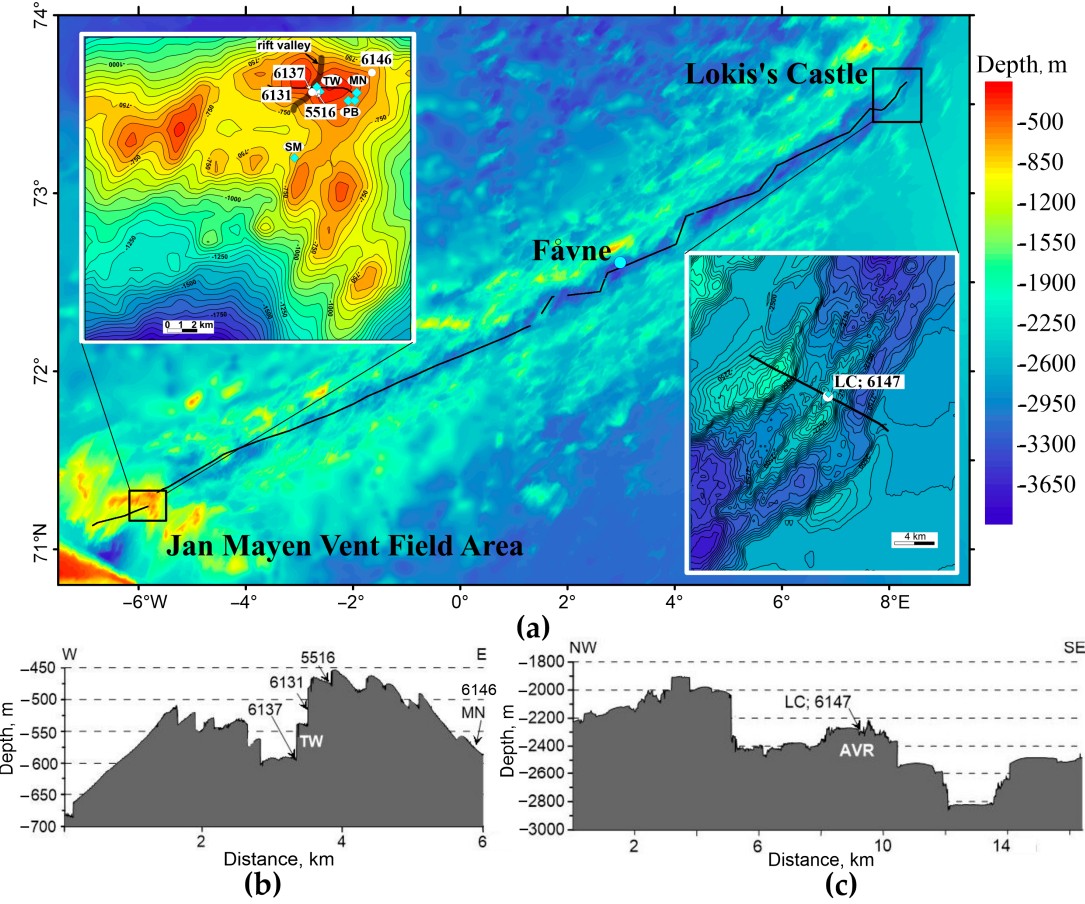

**Figure 2.** Bathymetric map of the Mohns Ridge (based on GEBCO bathymetry) and location of the hydrothermal vent fields (**a**); the upper left inset shows a bathymetric map of the Jan Mayen AVR top and location of the Soria Moria (SM), Gallionella Garden (GG), Troll Wall (TW), Perle and Bruse (PB) vent fields, and Fe-Mn crust area (MN) (filled blue circles); sampling stations considered in this article are shown by numbered filled white circles; the lower right inset shows a bathymetric map of the AVR with the Loki's Castle (LC) vent field. Bathymetric profile illustrating the morphology of the Jan Mayen AVR summit and the location of vent fields and stations of sampling (**b**); and a bathymetric profile through the AVR and the location of the Loki's Castle vent field and station of sampling (**c**).

### 3.1.1. Jan Mayen AVR at 71°18′ N and Related Hydrothermal Systems

The rift valley trends in the direction of 20–25°, dividing the Jan Mayen AVR summit into its NW and SE segments. The northwestern and southeastern walls of the rift valley are bounded by 2–3 scarps 40–75 m high. Scarps are 10–30 m high, oriented parallel to the rift valley, and are also observed on the NW and SE summit wings (Figure 2a, left inset, and Figure 2b). The scarps correspond to the normal faults that limit the steep sides of tilted blocks—structural elements typical for extension conditions.

The southeastern rift wall is occupied by the Troll Wall vent field and situated within a 150 m deep, 2 km wide, NE-SW-trending rift graben, which transects a large central Frøya Volcano [48]. The venting occurs through talus deposits at the base of the wall of a rift-bounding normal fault. Venting up to ~270 °C takes place at a relatively shallow (~550 m) depth and gives rise to numerous active white smoker chimneys [32]. Diffuse low-temperature hydrothermal fluids emanate from ridge parallel faults and fissures along the central rift floor, approximately 500 m west of the active relatively high-temperature venting area [55]. The temperature of the diffuse fluids is 2–7 °C above the ambient seawater temperature (–0.4 °C).

The Perle and Bruse vent field are located away from the rift valley on the flank of the SE segment of the AVR summit, which is cut by normal faults. An active outflow of fluid occurs along one of the normal faults parallel to the rift valley. That field is characterized by the presence of a hydroacoustic anomaly caused by the release of gas into the water column [34]. Unique deposits of Fe-Mn crusts were discovered and sampled within the Jan Mayen vent field area in the vicinity of the Perle and Bruse vent field (Figure 2a, left inset, and Figure 2b). The rare earth element distribution, positive Eu anomaly, high εNd values, and low $^{87}Sr/^{86}Sr$ ratios in the crust are indicative of the fact that the ore material of the crusts mainly originated from hydrothermal solutions [53]. The high-temperature endmember fluids, up to 242 °C, are venting through the hydrothermal chimneys. The diffuse venting (<54 °C) is emanating from the sediments surrounding the mound and is evidently derived from the Bruse endmember fluid [56].

### 3.1.2. The AVR at 73°30′ N and 8° E and Related Hydrothermal System

Loki's Castle vent field is located on the AVR and consists of hydrothermal debris from previously collapsed structures in the Mohns–Knipovich bend (Figure 1). This is the area of a sharp bend in the axial zone of the Mohns Ridge in the northern direction. The AVR's length is 30 km and its width is 4 km; it is located in the central part of the rift valley, the width of which is 8.5 km (Figure 2a, right inset).

The rift corresponds to a depression outlined with a contour line of 2300 m. Rift walls with a height of 50–100 m are distinguishable in the bathymetric profile obtained in the 75th cruise and crossing the Loki's Castle vent field (Figure 2c). The vent field consists of four active sulfide chimneys up to 13 m high, apparently associated with two sub-parallel normal faults that extend in a northeasterly direction and limit the rift's northwestern side [32,52]. The vent field is located on top of two sulfide mounds, forming a composite mound with a size comparable to that of the Trans-Atlantic Geotraverse (TAG) mound at 26°08′ N, MAR [37,57]. Even though the active chimneys have grown on a basaltic ridge, geochemical fluid data show a strong sedimentary influence on the hydrothermal circulation at Lokí's Castle [33] due to its close location to the Bear Island sediment fan. The hydrothermal field is venting up to 300–320 °C of hot black smoker fluids. Additionally, a low-temperature (~20 °C) vent area featuring numerous small barite-silica chimneys occurs adjacent to the eastern mound [13].

### 3.2. Field Observations and Sample Collection

Conductivity–temperature–depth (CTD) profiling using Seabird SBE911plus (Sea-Bird Electronics, Inc., Bellevue, WA, USA) was performed in order to document the ambient seawater properties. Buoyant plume particles were collected using cylindrical sediment traps deployed at a depth of 30 m from the seafloor [58].

Mineralized and hydrothermally altered sediments were collected by the grab ($0.25 M^2$ sampling area) from the Jan Mayen vent field area at 71° N within the Troll Wall and Perle and Bruse vent fields and the Loki's Castle vent field at 73° N, during the 68th and 75th cruises of the RV Akademik Mstislav Keldysh in July 2017 and June 2019 [54,59], respectively (Figure 2a). Six different sulfide-rich samples were investigated on board and immediately dried at 30 °C prior to processing.

The bathymetric and hydroacoustic investigations were performed with a Kongsberg EA 600 Series (Kongsberg Maritime AS, Horten, Norway) ship-mounted single-beam echo sounder operating at a constant frequency of 12 kHz. Windows XP®-based software (Microsoft, Redmond, WA, USA) is used to control the operation of the EA 600. The investigations were performed in the studied areas along separate transects.

### 3.3. Petrography and Mineralogy

All samples were examined using a Carl Zeiss Stemi 508 (Carl Zeiss SMT GmbH, Oberkochen, Germany) binocular microscope. The types of sediment were determined according to the Bezrukov and Lisitzin [60] classification of sea bottom sediments [61]. The samples were examined for purity by X-ray diffraction (XRD) and scanning electron microscopy (SEM) with energy-dispersive spectrometry (EDS). The composition of the sulfide-rich samples was studied in bulk and with hand-picked aggregates and grains. Sub-samples were finely crushed in an agate mortar, and the same 'bulk' powder was used for XRD.

Detailed mineralogical identification by XRD was performed on unoriented prepared specimens analyzed by a Bruker D8 Advance diffractometer (Bruker AXC, Karlsruhe, Germany) (Cu K$_\alpha$ with Ni 0.02 filter, 40 kV, 40 mA, with a linear detector LYNXEYE with scanning in a discrete mode with 0.02° steps, exposure 4 s/step in a range of 2.5°–70° 2$\theta$ range) in IO RAS. Mineral identification was performed with an automatic/manual peak search using Bruker's Diffrac EVA 3.1 (Bruker AXS, Karlsruhe, Germany). The PDF4 Mineral Database of ICDD and the Crystallography Open Database (COD, Vilnius, Lithuania) were used for identification purposes. Mineral quantification was performed by Rietveld refinement using the TOPAS 5.0 software (Bruker AXS, Karlsruhe, Germany). The list and abundance of hydrothermally derived minerals are represented in Table 1.

**Table 1.** List of identified hydrothermally derived minerals and their abundances in the studied assemblages from the sediments of the Troll Wall, Perle and Bruse, and Loki's Castle. Mineral abundance: major—identified in most samples examined, including bulk, fraction > 0.63 mm, and hand-picked aggregates; minor—identified regularly in hand-picked aggregates or grains; and rare—identified episodically in hand-picked aggregates or grains. Clay minerals are partly or mostly hydrothermal.

| Mineral | Formula | Abundance | Content Range, wt. % |
|---|---|---|---|
| | Troll Wall, proximal sediments | | |
| Smectite | $M_{x+y}^+(R_{2-y}^{3+}R_y^{2+})(Si_{4-x}\ Al_x)O_{10}(OH)_2 \cdot nH_2O$ * | Major | 15–62 |
| Illlite | $K_{0.6-0.85}(Al,Mg)_2(Si,Al)_4O_{10}(OH)_2$ | Major | 1–10 |
| Fe-Si oxyhydroxides | Fe amorphous silica | Major | n.d. ** |
| Barite | $Ba(SO_4)$ | Major | 3–85 |
| Gypsum | $CaSO_4\ 2H_2O$ | Major | 1–11 |
| Pyrite | $FeS_2$ (cubic) | Major | 4–99 |
| Marcasite | $FeS_2$ (orthorhombic) | Major | 4–87 |
| Sphalerite | $ZnS$ (cubic) | Major | 1–10 |
| Chalcopyrite | $CuFeS_2$ | Minor | 1–2 |
| Birnessite | $(Na,Ca)_{0.5}(Mn^{4+},Mn^{3+})_2O_4\ 1.5H_2O$ | Minor | n.d. |
| Buserite | $Na_4Mn_{14}O_{27}\ 21H_2O$ | Minor | n.d. |
| Asbolane–Buserite | Mixed-layer Mn formation | Minor | n.d. |
| Chlorite–smectite | Mixed-layer clay formation | Minor | 0–37 |
| Wurtzite | $(Zn)S$ (hexagonal) | Rare | 2–5 |
| Goethite | $\alpha$-FeO(OH) | Rare | 1–53 |
| Lepidocrocite | $\gamma$-FeO(OH) | Rare | 0–25 |
| X-ray amorphous clay | Ultrafine structurally defective clay formation | Rare | n.d. |
| Mordenite | $(Na_2,Ca,K_2)_4(Al_8Si_{40})O_{96}\ 28H_2O$ | Rare | 0–93 |
| Jarosite | $KFe_3^{3+}(SO_4)_2(OH)_6$ | Rare | 13 |
| Chlorite | $(Mg,Fe^{2+})_5Al(Si_3Al)O_{10}(OH)_8$ | Rare | n.d. |

**Table 1.** *Cont.*

| Mineral | Formula | Abundance | Content Range, wt. % |
|---|---|---|---|
| Perle and Bruse, distal sediments | | | |
| Smectite | $M_{x+y}^+(R_{2-y}^{3+}R_y^{2+})(Si_{4-x}Al_x)O_{10}(OH)_2 \cdot nH_2O$ | Major | n.d. |
| Illlite | $K_{0.6-0.85}(Al,Mg)_2(Si,Al)_4O_{10}(OH)_2$ | Major | 3–8 |
| Barite | $Ba(SO_4)$ | Major | 1–6 |
| Gypsum | $CaSO_4\ 2H_2O$ | Major | 1–6 |
| Birnessite | $(Na,Ca)_{0.5}(Mn^{4+},Mn^{3+})_2O_4\ 1.5H_2O$ | Major | n.d. |
| Buserite | $Na_4Mn_{14}O_{27}\ 21H_2O$ | Major | n.d. |
| Fe-oxides | X-ray amorphous $Fe_2O_3$ | Major | n.d. |
| Pyrite | $FeS_2$ (cubic) | Minor | 1–2 |
| Smectite–illite | Mixed-layer clay formation | Rare | 2–4 |
| Chlorite | $(Mg, Fe^{2+})_5Al(Si_3Al)O_{10}(OH)_8$ | Rare | 2–3 |
| Buserite | $Na_4Mn_{14}O_{27} \cdot 21H_2O$, heat resistant | Rare | n.d. |
| Asbolane | $(Ni, Co)_{2-x}Mn^{4+}(O, OH)_4\ nH_2O$ | Rare | n.d. |
| Asbolane–Buserite | Mixed-layer Mn formation | Rare | n.d. |
| Loki's Castle, proximal sediments | | | |
| Smectite | $M_{x+y}^+(R_{2-y}^{3+}R_y^{2+})(Si_{4-x}\ Al_x)O_{10}(OH)_2\ nH_2O$ | Major | 10–30 |
| Talc | $Mg_3Si_4O_{10}(OH)_2$ | Major | 1–95 |
| Pyrite | $FeS_2$ (cubic) | Major | 3–18 |
| Marcasite | $FeS_2$ (orthorhombic) | Minor | 3–30 |
| Pyrrhotite (hexagonal) | $Fe_{1-x}S$ | Major | 5–81 |
| Pyrrhotite (monoclinic) | $Fe_{1-x}S$ | Minor | 5–15 |
| Sphalerite | $ZnS$ (cubic) | Major | 2–13 |
| Chalcopyrite | $CuFeS_2$ | Major | 2–75 |
| Barite | $Ba(SO_4)$ | Major | 2–28 |
| Native sulfur | $S$ | Major | 4–34 |
| Goethite | $\alpha$-$FeO(OH)$ | Major | 3–18 |
| Lepidocrocite | $ɤ$-$FeO(OH)$ | Minor | 1–24 |
| Chlorite–smectite | Mixed-layer clay formation | Minor | n.d. |
| Talc-like mineral–smectite | Mixed-layer formation with Mg-Fe phyllosilicate | Minor | n.d. |
| Gypsum | $CaSO_4\ 2H_2O$ | Minor | 1–8 |
| Chlorite | $(Mg, Fe^{2+})_5Al(Si_3Al)O_{10}(OH)_8$ | Rare | n.d. |
| Galena | $PbS$ | Rare | 2–3 |
| Famatinite | $Cu_3SbS_4$ | Rare | 0–4 |
| Paratacamite | $Cu_3(Cu, Zn)(OH)_6Cl_2$ | Rare | 0–58 |
| Jarosite | $KFe_3^{3+}(SO_4)_2(OH)_6$ | Rare | 0–35 |

* Notes to the formula of smectite: $M^+$ indicates univalent cations; $R^{3+}$ indicates trivalent cations, usually represented by $Al^{3+}$ and $Fe^{3+}$; $R^{2+}$ indicates divalent cations, such as $Mg^{2+}$ and $Fe^{2+}$. ** n.d.—not detected.

Separate subsamples were examined using a VEGA-3sem TESCAN (Brno–Kohoutovice, Czech Republic) scanning electron microscopy (SEM) with an Oxford INCA Energy 350 (Oxford Instruments, Abingdon, UK) energy dispersive X-ray spectrometer (EDS). Polished thin sections from representative samples were selected for EDS analyses of hosted rocks. The EDS data is given in the Supplementary Material (Table S1). Barite grains and aggregates were selected from sediment samples for EDS analyses of specific mineral phases of barite.

*3.4. XRF Analysis*

X-ray fluorescence (XRF) analysis of 'bulk' powder sediment samples impacted by hydrothermal alteration and hyaloclastites was performed on a sequential spectrometer with wavelength dispersion, model Axios mAX (PANalytical, Eindhoven, Netherlands), and software SuperQ (PANalytical, Eindhoven, Netherlands). For the rock-forming oxide analyses, preparations were made from sample powder dried at 110 °C and melted with a mixture of lithium borates at 1150 °C. Glassy disks were formed from the resulting borate melt and analyzed on a spectrometer. To determine trace elements and sulfur, preparations were made from the sample powder by cold pressing with plastic filler at 30 tons pressure into 32 mm diameter tablets, which were analyzed on a spectrometer. The XRF major element composition of the sediment samples is given in the Supplementary Material (Table S2). The analysis was conducted in order to reveal the type of studied sediments and assess the metalliferous sediment index.

### 3.5. Sample Preparation and Fluid Inclusion Petrography

FI studies were performed using zoned barite crystals of 1000–100 µm in size (Figure 3a) collected from sulfide-rich assemblages from the Troll Wall (samples 5516, 6131, and 6137), Perle and Bruse (sample 6146), and Loki's Castle (samples 6147) hydrothermal sediments. Barite was separated from other minerals using the Olympus BX51 optical microscope. FIs used for measurements ranged in size from 5 µm to 15 µm in length. The most common size was between 8 µm and 15 µm. The microthermometric study was conducted on carefully selected FIs with no evidence of necking-down or leakage. At room temperature, all studied FIs contain a liquid phase and a vapor bubble.

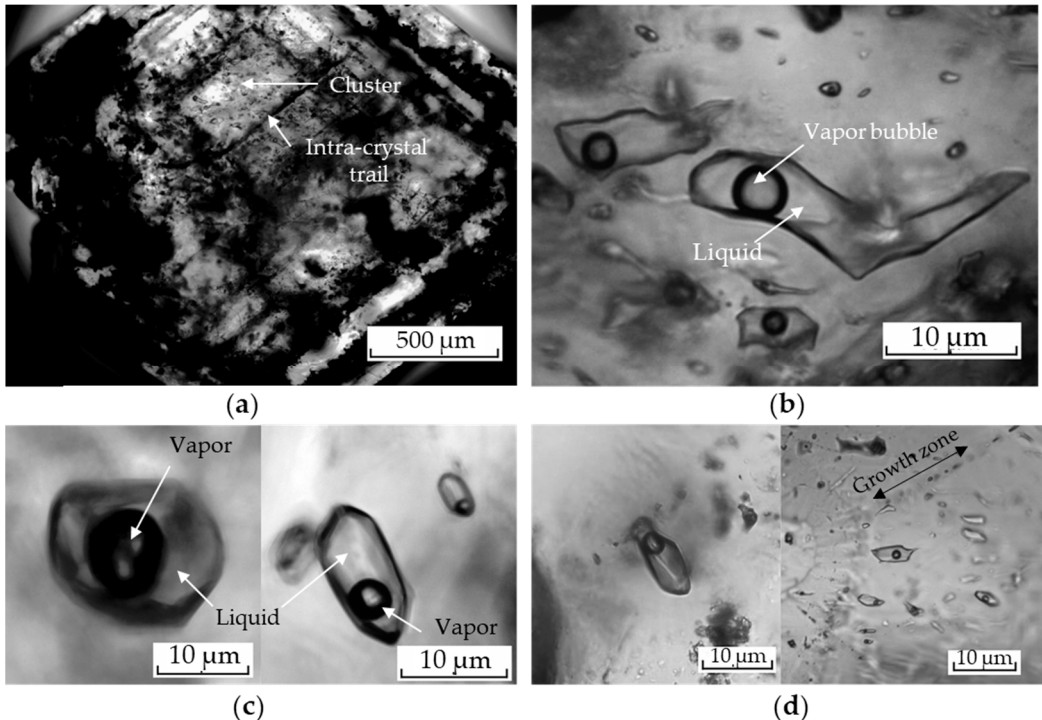

**Figure 3.** Microphotographs of zoned barite crystal (**a**) and two-phase liquid–vapor fluid inclusions in barite crystals (**b–d**).

Three textural generations of FIs were observed within the barite crystals: (1) presumed primary (P), (2) primary–secondary (P–S), and (3) secondary (S) inclusions. Primary two-phase liquid-vapor FIs occurring within barite crystals were just one type, with a vapor bubble occupying less than half the volume of the inclusion (volume of the vapor phase: 15–20 vol.%) and a liquid solution (Figure 3b–d). The observed FIs have a negative crystal shape. The primary origin of the FIs is determined according to the criteria proposed in Roedder [35]. Therefore, primary FIs are confined to growth zones or uniformly distributed in the host mineral volume; primary–secondary FIs are confined to fractures not extending beyond barite crystal boundaries; and secondary FIs are localized in fractures.

### 3.6. Fluid Inclusion Microthermonetry

The FIs studies were carried out at the Department of Geochemistry of the Institute of the Geology of Ore Deposits, Petrography, Mineralogy, and Geochemistry (IGEM), Russian Academy of Sciences, on a Linkam THMSG600 (Redhill, Surrey, UK) heating–freezing stage mounted on an Olympus BX51 (Tokyo, Japan) microscope and video camera. The above measuring setup allows real-time measurements to be made of the temperature of phase transitions in the inclusions in the range from −196 °C to 600 °C and to observe them at high magnifications and make digital microphotographs. Calibration of the heating–freezing stage was carried out using SYN FLINC® synthetic fluid inclusion temperature standards

(FLUID INC., Denver, CO, USA). Microthermometric measurements were calibrated against the $CO_2$ triple point and $H_2O$ (ice) melting points in the synthetic fluid inclusions, as well as their homogenization temperature.

The salt concentrations (salinities, reported as wt.% NaCl eq.) of liquid in FIs were calculated from the dissolution temperature of the ice crystal, using data from the work of [62]. The salt composition of the liquid in FIs was determined by eutectic temperatures, according to [63]. Salinities and liquid densities, as well as calculations of temperature corrections for pressure and trapping temperatures of FIs, were inferred using the software FLINCOR (University of Wisconsin, Madison, WI, USA) [64]. The $T_{ice\ melt}$ and $T_{hom}$ were determined with a precision of $\pm 0.1\ ^\circ C$ and $\pm 1\ ^\circ C$, respectively. Crystallization or trapping temperatures ($T_{cryst}$) were determined by applying a hydrostatic pressure correction to the homogenization temperatures ($T_{hom}$), which are defined as the minimum temperature at which a single-phase fluid is present in the inclusion. A pressure correction (P = 239 bar) was made assuming 2376 m water depth at Loki's Castle vent field, 1025 kg/m$^3$ seawater density, and the addition of 1 atm of air pressure above the sea. Consequently, the pressure correction for the Jan Mayen vent fields was 70 bar. The results of microthermometric measurements are given in Table 2.

**Table 2.** Fluid inclusion data for the Troll Wall, Perle and Bruse, and Loki's Castle seafloor sulfide deposits.

| Station, Depth, Layer | Type of FIs * | n | $T_{hom}$, °C | $T_{eut}$, °C | $T_{ice\ melt}$, °C | Salinity, wt.% NaCl eq. | $d$, g·cm$^{-3}$ | $T$ **, °C | $T_{cryst}$ ***, °C |
|---|---|---|---|---|---|---|---|---|---|
| | | | | Troll Wall vent field, depth of 512–600 m | | | | | |
| 6131 | P | 8 | 275 | −33 | −2.1 | 3.4 | 0.80 | 1 | 276 |
| 512 m | P | 4 | 246 | −31 | −2.3 | 3.8 | 0.84 | 3 | 249 |
| 0–20 cm | P | 4 | 218 | −33 | −2.5 | 4.1 | 0.88 | 3 | 221 |
| | P | 5 | 217 | −31 | −2.6 | 4.2 | 0.88 | 3 | 220 |
| | P−S | 3 | 195 | −31 | −2.5 | 4.1 | 0.90 | 3 | 198 |
| | P−S | 2 | 167 | −25 | −2.5 | 4.1 | 0.93 | 4 | 171 |
| | S | 2 | 115 | −27 | −2.3 | 3.8 | 0.98 | 4 | 119 |
| 6131 | P | 7 | 264 | −33 | −2.6 | 4.2 | 0.82 | 2 | 266 |
| 512 m | P | 6 | 234 | −32 | −2.1 | 3.4 | 0.85 | 3 | 237 |
| 1–9 cm | P | 5 | 234 | −32 | −2.1 | 3.4 | 0.85 | 3 | 237 |
| | P | 8 | 216 | −31 | −1.9 | 3.1 | 0.87 | 3 | 219 |
| | P | 6 | 215 | −34 | −2.1 | 3.4 | 0.88 | 3 | 218 |
| | P−S | 5 | 186 | −33 | −2.1 | 3.4 | 0.91 | 4 | 190 |
| | S | 3 | 156 | −34 | −1.9 | 3.1 | 0.94 | 4 | 160 |
| | S | 6 | 156 | −32 | −1.7 | 3.1 | 0.94 | 4 | 160 |
| 6137 | P | 8 | 234 | −33 | −2.5 | 4.1 | 0.86 | 3 | 237 |
| 600 m | P | 6 | 231 | −34 | −2.4 | 3.9 | 0.86 | 3 | 234 |
| 0–5 cm | P | 7 | 230 | −32 | −2.3 | 3.8 | 0.86 | 3 | 233 |
| | P | 4 | 223 | −30 | −1.9 | 3.1 | 0.86 | 3 | 226 |
| | P | 10 | 219 | −32 | −1.7 | 2.8 | 0.87 | 3 | 222 |
| | P | 6 | 218 | −30 | −2.2 | 3.6 | 0.87 | 3 | 221 |
| | P | 5 | 213 | −34 | −2.0 | 3.3 | 0.88 | 3 | 216 |
| | P | 7 | 211 | −33 | −2.2 | 3.6 | 0.88 | 3 | 214 |
| | P−S | 5 | 193 | −31 | −2.0 | 3.3 | 0.90 | 4 | 197 |
| | P−S | 5 | 166 | −32 | −2.1 | 3.4 | 0.93 | 5 | 170 |
| | P−S | 3 | 161 | −33 | −2.1 | 3.4 | 0.93 | 4 | 165 |
| | S | 4 | 159 | −30 | −1.8 | 3.0 | 0.93 | 4 | 163 |

**Table 2.** *Cont.*

| Station, Depth, Layer | Type of FIs * | n | $T_{hom}$, °C | $T_{eut}$, °C | $T_{ice\ melt}$, °C | Salinity, wt.% NaCl eq. | $d$, g·cm⁻³ | $T$ **, °C | $T_{cryst}$ ***, °C |
|---|---|---|---|---|---|---|---|---|---|
| 5516 | P | 3 | 260 | −35 | −1.9 | 3.1 | 0.81 | 2 | 262 |
| 540 m | P | 2 | 257 | −30 | −2.0 | 3.3 | 0.82 | 3 | 260 |
| 0–3 cm | P | 2 | 257 | −34 | −2.7 | 4.4 | 0.83 | 3 | 260 |
| | P | 16 | 247 | −33 | −2.0 | 3.3 | 0.83 | 3 | 250 |
| | P | 3 | 243 | −31 | −2.5 | 4.1 | 0.85 | 3 | 246 |
| | P | 5 | 243 | −32 | −1.9 | 3.1 | 0.84 | 3 | 246 |
| | P | 3 | 241 | −31 | −1.9 | 3.1 | 0.84 | 3 | 244 |
| | P | 2 | 240 | −32 | −1.6 | 2.6 | 0.84 | 3 | 243 |
| | P | 2 | 235 | −32 | −2.1 | 3.4 | 0.85 | 3 | 238 |
| | P | 4 | 226 | −32 | −2.2 | 3.6 | 0.86 | 3 | 238 |
| | P | 8 | 209 | −30 | −2.0 | 3.3 | 0.88 | 3 | 212 |
| | P | 3 | 203 | −32 | −2.1 | 3.4 | 0.89 | 3 | 206 |
| | P | 5 | 199 | −31 | −2.3 | 3.8 | 0.90 | 4 | 203 |
| | P | 5 | 198 | −33 | −2.6 | 4.2 | 0.90 | 4 | 202 |
| | P | 4 | 186 | −33 | −1.8 | 3.0 | 0.91 | 4 | 190 |
| | P | 2 | 128 | −30 | −1.8 | 3.0 | 0.96 | 4 | 132 |
| Range | | | 115–275 | −35−−25 | −2.7−−1.6 | 2.6–4.4 | 0.80–0.98 | 1–5 | 119–276 |
| Perle and Bruse vent field, depth of 620 m | | | | | | | | | |
| 6146 | P | 3 | 271 | −30 | −2.3 | 3.8 | 0.80 | 2 | 273 |
| 620 m | P | 4 | 266 | −34 | −2.4 | 3.9 | 0.81 | 2 | 268 |
| 0–9 cm | P | 6 | 263 | −32 | −2.3 | 3.8 | 0.81 | 2 | 265 |
| | P | 4 | 249 | −31 | −2.5 | 4.1 | 0.84 | 3 | 252 |
| | P | 3 | 248 | −31 | −2.1 | 3.4 | 0.83 | 3 | 251 |
| | P | 3 | 247 | −27 | −2.1 | 3.4 | 0.83 | 3 | 250 |
| | P | 6 | 238 | −34 | −2.4 | 3.9 | 0.85 | 3 | 241 |
| | P | 7 | 236 | −32 | −1.9 | 3.1 | 0.83 | 3 | 239 |
| | P−S | 4 | 225 | −33 | −2.3 | 3.8 | 0.87 | 3 | 228 |
| | P−S | 5 | 211 | −32 | −2.0 | 3.3 | 0.88 | 3 | 214 |
| | P−S | 2 | 184 | −30 | −2.6 | 4.2 | 0.92 | 4 | 198 |
| | S | 3 | 135 | −28 | −2.2 | 3.6 | 0.96 | 4 | 139 |
| Range | | | 135–271 | −34−−27 | −2.6−−1.9 | 3.1–4.2 | 0.80–0.96 | 2–4 | 139–273 |
| Loki's Castle vent field, depth of 2376 m | | | | | | | | | |
| 6147 | P | 13 | 287 | −32 | −2.6 | 4.2 | 0.78 | 20 | 307 |
| 2376 m | P | 4 | 281 | −35 | −1.8 | 3.0 | 0.77 | 20 | 301 |
| 0–12 cm | P | 7 | 273 | −32 | −2.2 | 3.6 | 0.79 | 20 | 293 |
| | P | 2 | 265 | −31 | −2.7 | 4.4 | 0.82 | 20 | 285 |
| | P | 4 | 261 | −33 | −2.1 | 3.4 | 0.81 | 20 | 281 |
| | P | 5 | 256 | −34 | −1.9 | 3.1 | 0.82 | 20 | 276 |
| | P | 6 | 253 | −33 | −1.8 | 3.0 | 0.82 | 20 | 273 |
| | P | 2 | 251 | −33 | −2.3 | 3.8 | 0.83 | 20 | 271 |
| | P | 3 | 244 | −30 | −2.3 | 3.8 | 0.84 | 19 | 263 |
| | P | 9 | 234 | −35 | −2.6 | 4.2 | 0.86 | 19 | 253 |
| | P | 6 | 228 | −34 | −2.1 | 3.4 | 0.86 | 19 | 247 |
| | P−S | 2 | 227 | −30 | −1.9 | 3.1 | 0.86 | 19 | 246 |
| | P−S | 6 | 219 | −31 | −2.4 | 3.9 | 0.88 | 18 | 237 |
| | P−S | 7 | 218 | −31 | −2.1 | 3.4 | 0.87 | 18 | 236 |
| | P−S | 5 | 214 | −33 | −2.2 | 3.6 | 0.88 | 18 | 232 |
| | P−S | 7 | 205 | −32 | −1.7 | 2.8 | 0.88 | 18 | 223 |
| Range | | | 205–287 | −35−−30 | −2.7−−1.7 | 2.8–4.4 | 0.77–0.86 | 8–20 | 223–307 |

* Genetic types of FIs: P—primary, P–S—primary–secondary, and S—secondary. n—number of FIs. ** Pressure correction: Troll Wall—~59 bar, Perle and Bruse—61 bar, and Loki's Castle—233 bar. *** $T_{cryst}$ of hosted mineral (fluid trapping temperature).

## 4. Results

### 4.1. Temperature and Salinity of Ambient Seawater

Most of the water column from the seafloor to a layer of ~140 m in the Jan Mayen vent field area is occupied by cold Arctic intermediate water with temperatures and salinities of $-0.27$–$0.3$ °C and 34.88–34.91 psu, respectively. Only the upper quasi-uniform layer of about 0–60 m is warmed to 2–3 °C. The measured ambient seawater temperature varied from $-0.27$ °C to $-0.19$ °C and from $-0.13$ °C to $-0.10$ °C at different depth ranges of 774–660 m and 600–550 m, respectively. Positive temperature anomalies were recorded within the buoyant hydrothermal plumes, where the seawater temperature reached 0.64–2.49 °C.

In the area of Loki's Castle vent field, the deep-water mass of ~1500 m thickness is occupied by cold Arctic intermediate water with sub-zero temperatures and a salinity of 34.91 psu. The upper quasi-uniform layer (0–60 m) is warmed to 5.59 °C. The measured ambient seawater temperature reaches $-0.72$ °C at a depth of 2368 m. Therefore, the ambient temperatures varied from $-0.10$ °C to $-0.72$ °C and depended on the depth of the vent fields. The ambient salinity is approximately 34.91 psu at both locations, which is 31.85 psu NaCl eq.

### 4.2. Petrography of Basaltic Rocks and Lithology of Hydrothermal Sediments

Mineralized and hydrothermally altered sediments of variable thickness rest on a substrate composed primarily of silicified hyaloclastite and basaltic debris. Boulders of hyaloclastites and basalts, as well as breccia, are widespread on the seafloor of the Jan Mayen vent field area. Volcanic tuffs (~15 × 10 × 5 cm) of dark brown to black colors were collected in this area, containing on the surface fragments of shells, glandular crusts, and organic matter residues.

Hydrothermal sediments of the Troll Wall vent field are coarse- and medium-grained dark gray tuffaceous sands with inclusions of basalt lithoclasts (medium and fine rubble), ferruginous tuffs, and psammite, siltstone, and mudstone breccia sized 1–8 cm, often with a ferrous film on the surface. Sands contain lapilli-size volcanic fragments, bright brown ferruginous crust-like fragments, red spots of ferrugination at the surface, and characteristic signs of hydrothermal alterations in the form of white and gray fluffy clay. Fragments of rocks, shells, and pebbles (4 × 3 cm) are present as well. Beneath the thin (~1 cm) oxidized surface layer of the sediment, a patchy sediment texture with reddish-gray compaction appears, probably along zones of diffuse fluid penetration. The sands are weakly iron-rich (5 to 10 wt.% Fe, according to [61]), with an average Fe content of 8.0 ± 0.9 wt.% in the studied samples.

Hydrothermal sediments of the Perle and Bruse vent field are medium- and fine-grained tuffaceous dark grayish-brown sands with basalt inclusions (fine rubble) and psammite structure. Beneath the thin (~1 cm) oxidized surface layer, a spotted texture appears on the background of a very dark grayish-brown or brown sediment. The influence of normal pelagic marine sediments at the Perle and Bruse vent field is more significant than at the Troll Wall vent field. In terms of iron content, the sands are usually weakly iron-rich, with an average Fe content of 6.0 ± 0.1 wt.%. Nonetheless, ferruginous (10–20 wt.% Fe) sands with an average Fe content of 14.5 ± 0.2 wt.% and weakly manganese (5–0.2 wt.% Mn) sands with an average Mn content of 0.35 ± 0.1 wt.% were also collected in this area.

Hydrothermal sediments of the Loki's Castle vent field are coarse, unsorted muddy sands with a psammite structure. The top layer (0–6 cm) is brown sand with ochreous, yellow, and black grains, ochreous crust-like agglomerates up to 3 cm, and white fluffy clay on the surface of the agglomerates. Deeper (6–12 cm) are oblique layers of dark reddish-brown sediment, as well as areas of very dark grayish-brown sediment. Close to the base of the exposed sand layer, dense agglomerates are often covered with cubic pyrite crystals. The texture of the sediment is pseudo-agglomerate, spotty-layered, and heterogeneous with sharp boundaries. Fluid conduits were observed in a regular circular shape up to ~10 cm in

diameter with sharp boundaries and denser bluish-gray sediment inside. There were fine quartz and feldspar grains, obviously ice-rafted debris, coated with an iron film. In terms of iron content, the sands are weakly iron-rich (9 wt.% Fe) and ferruginous (>10 wt.% Fe).

Basaltic rocks are present in hydrothermal systems such as the Jan Mayen and Loki's Castle vent fields. On the rift wall occupied by Troll Wall, volcanic tuff and semi-lithified volcaniclastic psammite breccia (more than 70% volcanic glass) were mostly collected. Volcanic breccia is polymictic of dark gray and bluish color with ochre-colored and black-colored interlayers and contains fine pebbles and gravel and inclusions of gray silt nodules. The volcanic breccia often has signs of hydrothermal alteration and mineralization: sulfide crystals (dominantly cubic pyrite), barite, and Fe-Si oxyhydroxides. The basaltic and glassy fragments are widespread in the surface sediments of the Troll Wall and Perle and Bruse vent fields. Basalt lithoclasts of wide size (till 5 cm) and often angular and subangular shape are characterized by a vesicular structure with porphyritic and microporphyritic texture, dominated by volcanic glass and, in smaller amounts, containing split phenocrysts of plagioclase and idiomorphic phenocrysts of clinopyroxene and olivine (Figure 4). The volcanic glass in these samples differs in the degree of alteration: from colorless, transparent, and dense, to light yellow and fractured, with areas of palagonite (altered glass) and smectite, to a black isotropic opaque mass composed entirely of smectite microfibers.

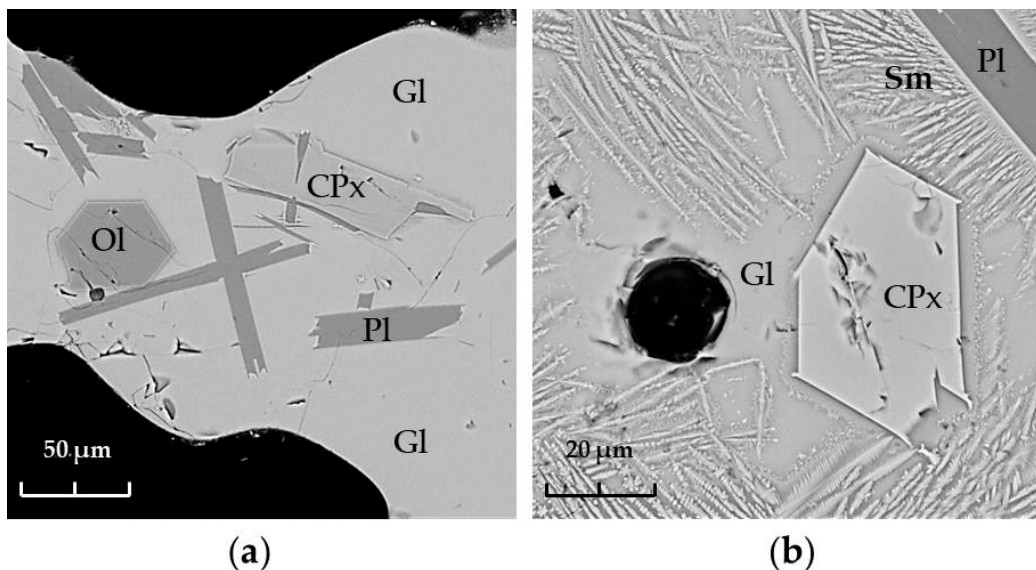

**Figure 4.** SEM micrographs of volcanic glasses with different degrees of alteration: (**a**) fragment of the glass with phenocrysts of plagioclase, clinopyroxene, and olivine, indicating very rapid cooling; (**b**) fragment of the glass with phenocrysts and hydrous layer silicates such as smectite-group minerals. Abbreviations: Gl—glass, CPx—clinopyroxene, Pl—plagioclase, Ol—olivine, Sm—smectite.

Glassy pillow basalts and occasional glass fragments were detected in the surface sediments of the Loki's Castle vent field. Basalt lithoclasts are characterized by a sparsely phyric texture, dominated by volcanic glass, small amounts of plagioclase and olivine phenocrysts, and less frequently pyroxene phenocrysts. Yellow and light brown glass are characterized by altered areas with palagonite and ochre microfibers. Coarse (1–4 cm) angular shards of black glass, usually with ochre film, were collected from the surface of soft sediments in the proximal area of the vent field. The isotropic opaque glass shards demonstrate curving fracture rims, typical of conchoidal fracture.

Hydrothermal crust-like orange-ochre, yellow-ochre, yellow, dark red, and black agglomerates occur in the volcanic breccia and sediments of the studied Jan Mayen vent fields. The matrix of the ochre-colored crusts is represented by collomorphic Fe-oxides, yellow-colored or dark red—usually by Fe–Si oxyhydroxides, and black—by collomorphic Fe and Mn oxyhydroxides. The newly discovered Fe-Mn crusts in the distal area of the

Perle and Bruse vent field [53] are layered slabs (up to ~3 cm thick and ~25 × 12 cm in size) composed of a homogeneous ore matrix of Mn and Fe oxyhydroxides with an admixture of non-metallic matter, mainly fine-dispersed volcanic glass. The ore matrix of the crusts consists mainly of Mn hydroxides with a subordinate admixture of Fe oxides. Crust-like agglomerates, grains, and globules of ochreous, yellow, and black colors, represented by Fe-oxides, talc, and native sulfur, are found in the sediments of the Loki's Castle vent field.

### 4.3. Mineralogy of Hydrothermal Sediments

XRD data of the Troll Wall sediments indicates plagioclase and smectite (hereinafter following in decreasing order of content) as dominant phases in bulk mineralogy, followed by pyroxene and olivine. Quartz, feldspar, and other clay minerals (kaolinite, chlorite, and illite) occur much less frequently or in trace amounts. On the other hand, mixed-layer clays with swelling layers of the chlorite–smectite type are common in sediments. It has been found that X-ray amorphous clay may be present in the surface sediment layer. Sulfides are represented mostly by pyrite and marcasite and less frequently by sphalerite, chalcopyrite, and wurtzite. Cu and Zn sulfides are often precipitated as collomorphic textures of sphalerite and chalcopyrite in association with barite. The hydroxide class is mainly represented by Fe-oxides, such as goethite and lepidocrocite, and Mn-hydroxides, such as birnessite, buserite I, and asbolane-buserite, as subordinate minerals. Fe–Si oxyhydroxides (halo in the angular interval 17–30 2$\Theta$° with a maximum in the range of approximately 4 Å) are found in sediments and form edifices. Mordenite-type zeolite was detected in the altered volcaniclastic breccia. Among the sulfate minerals, barite was identified as well as gypsum and jarosite as accompanying minerals. The sediments are rich in barite (till 19 wt.% in bulk mineralogy and till 85 wt.% in aggregates associated with sphalerite), which forms mineral associations with sulfides.

XRD data from the Perle and Bruse sediments indicate calcite, plagioclase, and quartz as dominant phases in bulk mineralogy, which are followed by pyroxene. Feldspar and clay minerals (kaolinite, chlorite, illite, and minor smectite) occur much less frequently or in trace amounts. Mixed-layer clays with swelling layers of the smectite–illite type are also identified in sediments. Sulfides are represented mostly by pyrite. The hydroxide class is dominated by Mn-hydroxides, such as birnessite, buserite I, and asbolane or asbolane–buserite, as subordinate minerals. Fe oxides are common but do not form their own crystalline mineral phase in sediments. Barite and gypsum are identified among the sulfate minerals (1–6 wt.% in total).

The XRD data of the Loki's Castle sediments indicates that the dominant phases are smectite, talc, and quartz, followed by plagioclase and calcite. Other clay minerals (kaolinite, chlorite, and illite) occur in trace amounts. The sediments reveal mixed-layer clays (mostly hydrothermal) with swelling layers of chlorite-smectite type and talc-like mineral-smectite type, which according to [38] has been interpreted as talc-like kerolite-smectite of Mg-Fe phyllosilicate type. The former contains fine sphalerite and goethite; the latter contains fine sulfides such as hexagonal pyrrhotite, chalcopyrite, and sphalerite. Sulfides are represented by a wide range of mineral varieties: pyrite, pyrrhotite (monoclinic and hexagonal), chalcopyrite, sphalerite, and less frequently marcasite, galena, and famatinite. The content of sulfide minerals varies from 3 wt.% to 72 wt.% in total mineralogy and different rock fragments of sediments, respectively. Native sulfur is common in studied sediments, most often in the form of globules or as part of aggregate clusters. The hydroxide class is mainly represented by Fe-oxides, such as goethite and lepidocrocite. The Cu-Zn chloride mineral paratacamite occurs as dark aggregates in association with lepidocrocite and goethite. Among the sulfate minerals, barite, jarosite, and gypsum were identified. The sediments are rich in barite (2–15 wt.% in bulk mineralogy), which forms mineral associations with sulfides. Dark-colored fractions < 63 µm are rich in barite up to 28 wt.%.

### 4.4. Wholerock Composition and Dominant Trace Elements

To reveal the possibility of a non-detrital origin for Si content in the studied sediments, Si was plotted against Al (Figure 5), according to Turekian et al. [65]. There is no pairwise correlation between Si and Al in these sediments due to the high contribution of basalts. The Si/Al ratio in the sediments varied from 1.3 to 3.9, and in some of the samples it was around 3.0, indicating the input of detrital material (Figure 6a). Perle and Bruse sediments have slightly increased Si/Al ratios: ~3.5 on average. The Si/Al ratio in Loki's Castle sediments reached 25, confirming the higher Si value and reflecting the high clay component and the igneous mafic nature.

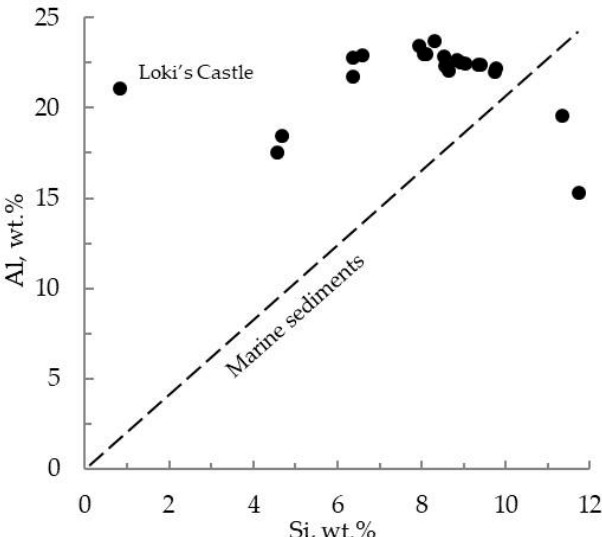

**Figure 5.** Si versus Al content in the hydrothermally altered sediments. The dotted line indicates values for pelagic marine sediments.

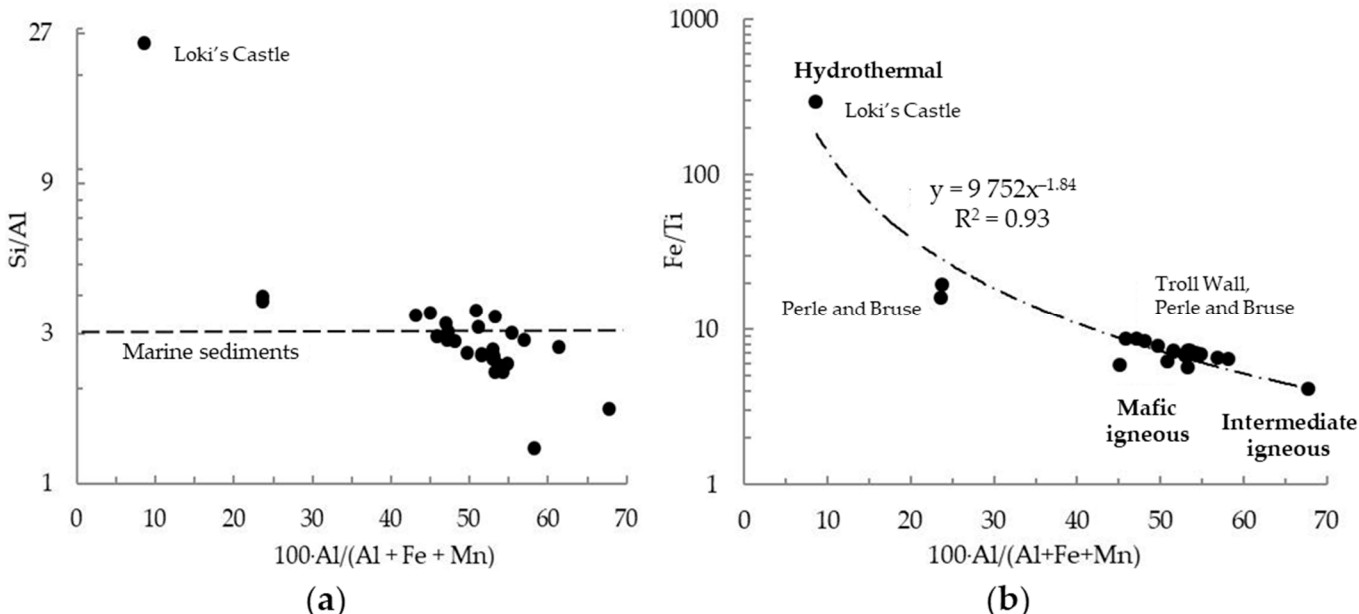

**Figure 6.** Plot of Si/Al ratio (**a**) and Fe/Ti ratio (**b**) versus metalliferous sediment index after [66] for the hydrothermally altered sediments. The y axis in both plots is a logarithmic scale. The dotted line in (**a**) indicates values for pelagic marine sediments.

All sediment samples examined have Mn/Fe ratios < 0.03 and cannot be classified as metalliferous sediments according to [67], where this ratio must exceed 0.07. The Perle

and Bruse sediment samples demonstrate Mn/Fe values of 0.03–0.02 indicative of little diagenetic alteration [68]. The values of Boström et al.'s [66] metalliferous sediment index, $100 \cdot Al/(Al + Fe + Mn)$, for Loki's Castle and Perle and Bruse sediments reached 8.7 and 23.7, respectively. Therefore, the extremely Al-depleted (0.8 wt.%) and Mg-enriched (14 wt.%) sediments of Loki's Castle reflect the higher Fe content due to stronger hydrothermal alteration with the formation of sulfides and talc. Al-depleted (4.6 wt.%) and Fe-Mn-enriched sediments within the Perle and Bruse vent fields are approaching distal metalliferous sediments, which reflects the high Fe and Mn oxyhydroxide input. Other Perle and Bruse sediments and all studied Troll Wall sediments demonstrate the variation of the metalliferous sediment index between 45.8 and 67.8, reinforcing the dominance of the detrital character of the sediments. The majority of samples of sediments from the Perle and Bruse and Troll Wall vent fields exhibit elements indicative of a predominantly igneous mafic and intermediate component, while the Loki's Castle sediments show a marked influence of a hydrothermal component (Figure 6b).

Ba content in Troll Wall and Perle and Bruse sediments varied from 388 ppm to 8800 ppm and reached 16,500 ppm in Loki's Castle sediments. Ba versus $S_{total}$ demonstrate reliable linear approximation (R = 0.95, n = 23) for studied sediments, indicative of a dominant hydrothermal affinity (Figure 7). $S_{total}$ content reaches 2.6 wt.% and 9.8 wt.% in the sediments of Loki's Castle and Troll Wall, respectively.

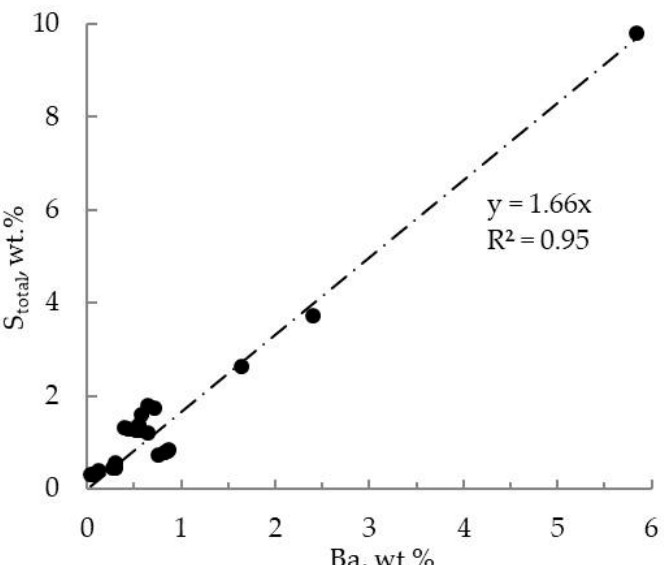

**Figure 7.** Ba versus $S_{total}$ content in the hydrothermally altered sediments.

In addition to Fe and Mn, three main metals are indicative of hydrothermal alteration: Cu, Zn, and Pb. Cu content varied from 58 ppm (which is close to its content in glassy fragments, from 55 to 71 according to Elkins et al. [51] and this work, respectively) to 1110 ppm in sediment samples of the Troll Wall and Perle and Bruse vent fields. Zn content varied from 57 ppm (which is depleted compared to the average content in glassy fragments, from 104 ppm to 83 ppm according to Elkins et al. [51] and this work, respectively) to 3140 ppm in sediment samples of the Troll Wall and Perle and Bruse vent fields. The higher Cu and Zn contents are characteristic of the Troll Wall sediments. In Loki's Castle sediment samples, the content reaches 3600 ppm and 4800 ppm for Cu and Zn, respectively. Pb content varies from 18 ppm (which is close to its content in pelagic sediments of 16 ppm [38]) to 109 ppm in sediment samples of the studied Jan Mayen vent fields, while its content in basalts is approximately 2 ppm [51]. In Loki's Castle sediment samples, the Pb content reaches 2200 ppm. As and Se are also noteworthy. In Perle and Bruse sediment samples, the As content reaches 211 ppm. In Loki's Castle sediment samples, Se content is as high as 45 ppm.

### 4.5. Barite Chrystal Morphology, Size, and EDS Data

Barite crystals have been found in buoyant plume particles and in sediments from both the Jan Mayen and Loki's Castle vent fields. Flat, micron-sized barite crystals with a crosscutting tabular shape of hexagonal type are identified in the buoyant plume extracted from sediment trap samples (Figure 8a–c). The tabular crystals form rosette structures in both hydrothermal settings, but crystal and rosette sizes vary. In the Jan Mayen vent fields, crystals >10 µm form well-shaped rosettes up to approximately 50 µm and are coarse. In Loki's Castle vent field, crystals of 5–10 µm form rosette-like structures up to approximately 15–20 µm. Barite forms mineral associations and aggregates with Fe, Cu, and Zn sulfides. Sulfide minerals form porous masses as bud-shaped masses with individual spherulites of <0.5 µm in size, and only pyrite forms regular octahedrons and cuboctahedrons up to 3 µm in size.

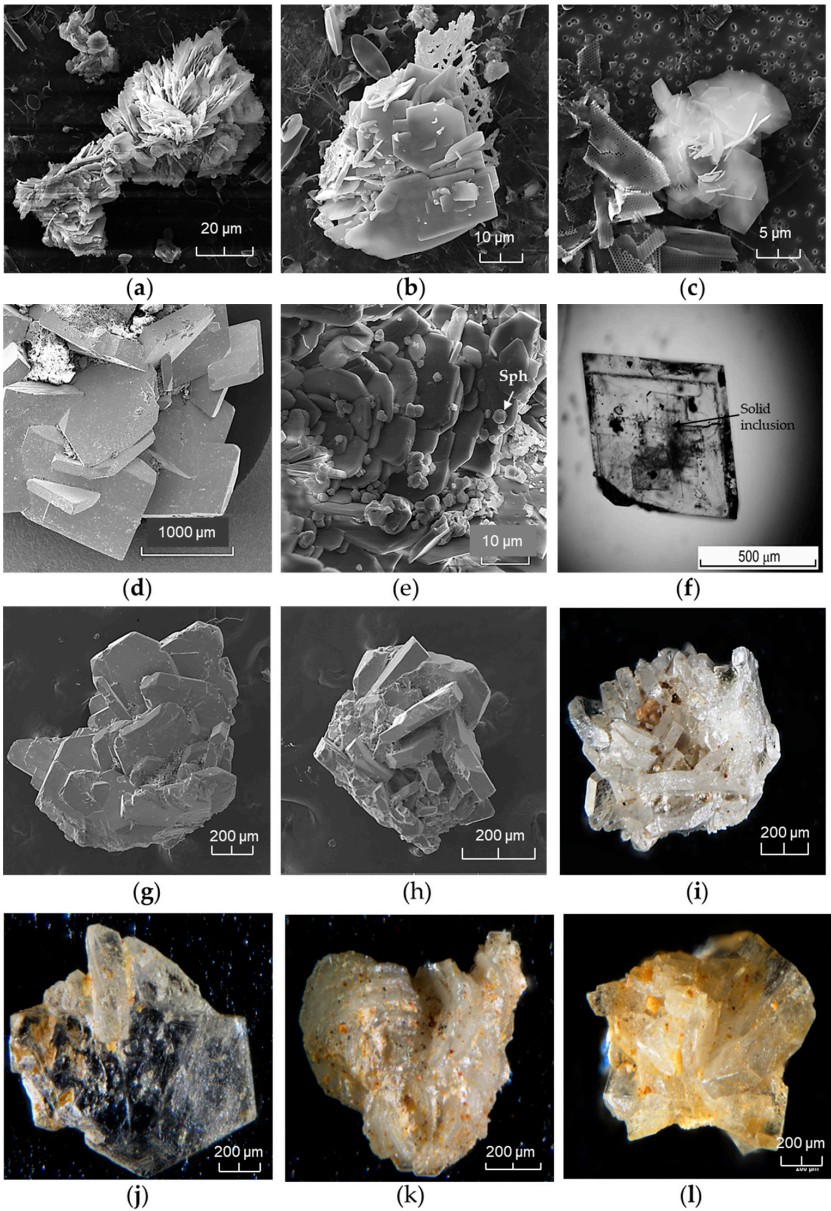

**Figure 8.** Scanning electron micrographs (**a**–**e**,**g**,**h**) and micrographs in plane polarized light (**f**,**i**–**l**) of the barite crystals: (**a**,**b**) sinking particles from buoyant plumes of the Troll Wall vent field; (**c**) sinking particles from buoyant plumes of the Loki's Castle vent field; (**d**–**f**) crystals from the surface layer of sediments of the Troll Wall vent field; and (**g**–**l**) crystals from the surface layer of sediments of the Loki's Castle vent field. Note: Sph—sphalerite.

In sediments, flat crystals are larger, approximately 20–1000 μm, and are typically precipitated in tabular shapes of hexagonal and rectangular types (Figure 8d–l). Polysynthetic twinning is often found. Idiomorphic crystals typically form rosette and druse structures up to approximately 3 mm and are also found as individual crystals. There are colorless and water-clear crystals, as well as white with cloudy core crystals and dark gray and black impurities. Yellow crystals occur, probably containing Fe oxides. Two generations of barite are usually distinguished, which form mineral associations with Fe, Cu, and Zn sulfides. Fine barite-I crystals form associations with sulfides where pyrite is the dominant mineral. Coarse barite-II crystals have formed as a result of free growth in voids in association with sulfides dominated by marcasite. Barite-II crystals are sector-zoned and contain numerous liquid and vapor FIs. The latter, tabular barite of the rectangular type, was usually used for FI investigations (Figure 8f). Coase barite crystals are characteristic of the Jan Mayen sediments, while smaller ones prevail in the Loki's Castle sediments.

The substitution of cations other than $Ba^{2+}$ into barite is identified. Minor substitution of $Ba^{2+}$ by $Sr^{2+}$, followed by $Ca^{2+}$, and $Fe^{2+}$ occurs in the studied barites (Table 3). Negligible substitution of $Ba^{2+}$ by $Co^{2+}$ occurs only in Loki's Castle, where high whole-rock Co concentrations were previously determined [37] as simple substitutions (e.g., spin state compatibility between $Co^{2+}$ and the divalent cation).

**Table 3.** EDS data of barite crystals (wt.%) collected from sediments and buoyant plumes at the Troll Wall (station 5516) and Loki's Castle (stations 6147 and 7049) vent fields.

| Grain | Ba | Sr | Ca | Fe | Co | S | O | Total |
|---|---|---|---|---|---|---|---|---|
| Hydrothermally altered sediments of Troll Wall | | | | | | | | |
| 1 | 57.60 | 1.99 | 0.26 | 0.44 | n.d. | 14.68 | 24.53 | 99.50 |
| 2 | 55.58 | 3.20 | 0.21 | 0.71 | n.d. | 15.14 | 26.53 | 101.37 |
| 3 | 57.69 | 1.33 | n.d. * | 0.51 | n.d. | 14.62 | 25.52 | 99.68 |
| 4 | 57.19 | 2.03 | n.d. | 0.93 | n.d. | 15.13 | 25.95 | 101.24 |
| 5 | 56.03 | 2.79 | n.d. | 0.60 | n.d. | 14.84 | 23.80 | 98.06 |
| 6 | 58.28 | 1.18 | n.d. | 0.57 | n.d. | 14.66 | 25.47 | 100.15 |
| Mean | 57.06 | 2.09 | 0.24 | 0.63 | - | 14.85 | 25.30 | 100.16 |
| Buoyant hydrothermal plume of Troll Wall | | | | | | | | |
| 7 | 66.78 | 1.70 | 0.40 | n.d. | n.d. | 14.41 | 16.71 | 100.00 |
| 8 | 50.10 | 1.41 | 0.49 | n.d. | n.d. | 13.87 | 30.67 | 96.54 |
| 9 | 41.49 | 5.32 | 1.30 | n.d. | n.d. | 15.89 | 35.25 | 99.25 |
| Mean | 52.79 | 2.81 | 0.73 | - | - | 14.72 | 27.54 | 98.59 |
| Hydrothermally altered sediments of Loki's Castle | | | | | | | | |
| 10 | 63.36 | 0.59 | 0.05 | 0.16 | 0.28 | 13.86 | 29.26 | 107.55 |
| 11 | 61.49 | 0.43 | 0.18 | 0.01 | 0.08 | 15.48 | 23.67 | 101.34 |
| 12 | 61.37 | 0.22 | n.d. | 0.10 | n.d. | 13.69 | 19.53 | 94.91 |
| 13 | 61.32 | n.d. | 0.15 | 0.30 | 0.24 | 14.15 | 24.15 | 100.31 |
| 14 | 63.24 | n.d. | 0.03 | 0.31 | n.d. | 15.09 | 23.38 | 102.05 |
| 15 | 59.49 | 0.03 | 0.01 | 0.17 | 0.03 | 16.85 | 29.37 | 106.95 |
| 16 | 57.76 | 0.94 | 0.10 | 0.29 | n.d. | 13.25 | 30.20 | 102.53 |
| Mean | 61.15 | 0.44 | 0.09 | 0.19 | 0.16 | 14.62 | 25.65 | 102.23 |
| Buoyant hydrothermal plume of Loki's Castle | | | | | | | | |
| 17 | 50.16 | 5.07 | 1.54 | n.d. | 0.04 | 14.37 | 29.88 | 99.96 |

\* n.d.—not detected.

### 4.6. Study of Fluid Inclusions

The microthermometric data were measured from two-phase FIs. Single-phase inclusions (liquid only) are not present in the crystals studied. The presence of only one type of FI determines the absence of FI assemblages (coexistence of FI of different types) in the studied barites.

The results of thermal and cryometric studies of more than 200 individual FIs in barite from hydrothermally altered sediments (Table 2, Figure 3) demonstrate that Fe, Mg, and Na chlorides prevailed in the hydrothermal fluids, as evidenced by the eutectic fluid temperature from −25 °C to −35 °C. Ice-melting temperatures ranged from −1.7 to −2.7 °C, corresponding to salinities between 2.8 and 4.4 wt.% NaCl eq. The relatively high temperature of ice melting indicates that the fluid trapped in the FIs was diluted with seawater.

In the Troll Wall vent field, the two-phase FIs homogenize at temperatures of 115–275 °C when heated. At the same time, the salt concentration and fluid density are 2.8–4.2 wt.% eq. NaCl and 0.80–0.98 g/cm$^3$, respectively. In the Perle and Bruse vent field, the homogenization temperature of two-phase FIs in barites is 159–234 °C, the salt concentration is 3.1–4.2 wt.% eq. NaCl, and the fluid density is 0.80–0.96 g/cm$^3$. In the Loki's Castle vent field, the homogenization temperature of two-phase FIs in barites is 205–287 °C, the salt concentration is 2.8–4.4 wt.% eq. NaCl, and the fluid density is 0.78–0.88 g/cm$^3$. However, to estimate the barite crystallization temperature through the homogenization temperature of the FIs, a correction for the pressure corresponding to the sampling depth is necessary. Due to the different depths at the studied hydrothermal vent fields, the values of temperature pressure correction are 1–5 °C and 18–20 °C for the Jan Mayen and Loki's Castle vent fields, respectively (Table 2).

Consequently, the data suggest that studied barites from the Troll Wall vent field crystallized at pressure corrected trapping temperatures of at least 276 °C from a heated fluid with salinity up to 4.4 wt.% eq. NaCl. Barites from the Perle and Bruse vent field crystallized at similar fluid temperatures and salinities: at least 273 °C and 4.2 wt.% eq. NaCl. The Loki's Castle hydrothermal fluid reached at least 307 °C and 4.4 wt.% NaCl eq. (Figure 9). Based on the salinity of the fluid, it could have been heated ambient seawater with a temperature of slightly below zero (from −0.10 °C to −0.72 °C) and a salinity of 3.185 wt.% NaCl eq. The data also reveal a general dilution of hydrothermal fluid down to 2.8–3.1 wt.% NaCl eq.

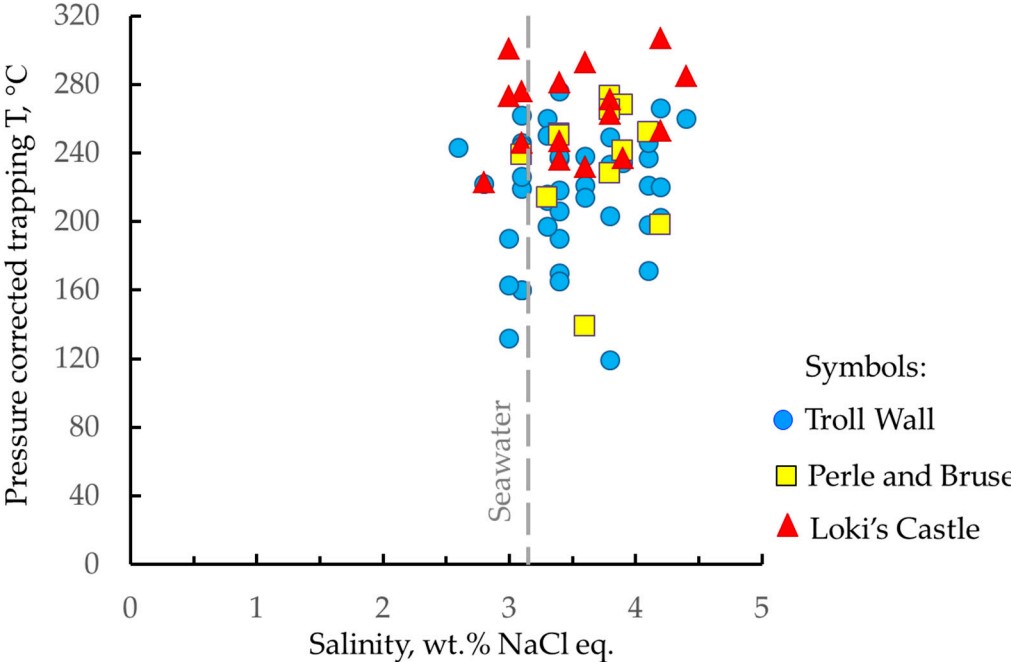

**Figure 9.** Fluid inclusion data hosted in barites for the Troll Wall, Perle and Bruse, and Loki's Castle hydrothermally altered sediments. Numerical data can be found in Table 2. A gray dotted line indicates the salinity of ambient seawater.

## 5. Discussion

The studied hydrothermal systems are associated with modern submarine volcanism and located at the AVRs, according to published [69] and own data. The Troll Wall is hosted by the SE rift wall of the AVR, which is characteristic of the MAR hydrothermal systems [40]. The Perle and Bruse vent field occupy the rift valley on the SE flank of the AVR summit, which is not a typical location for hydrothermal systems. The area of the vent field is exposed to detrital sediment and could be partially covered by a thin layer (2–3 cm) of pelagic clastic sediment. The sediment-influenced Loki's Castle vent field is located near the summit of the AVR. Although the active chimneys are on a basalt ridge, Baumberger et al. [33] clearly demonstrated the strong sedimentary influence on the hydrothermal circulation at Loki's Castle. Focused and diffuse venting occurs in the studied hydrothermal fields [13,32,56] that form volcanogenic massive sulfide deposits [37,40].

### 5.1. Characterization of the Bottom Water Layer at the Hydrothermal Vent Fields

The ambient seawater of both sites is characterized by temperatures slightly below zero and a salinity of 3.185 wt.% NaCl eq. The CTD data let us reveal positive temperature anomalies with amplitudes up to 2.49 °C in the near bottom layer associated with the propagation of a neutral buoyant plume in the Jan Mayen vent field area. In the near-bottom layer, water transport in a northeasterly direction was recorded, almost coinciding with the direction of the rift valley. The water movement was back-and-forth with a semidiurnal tidal cycle. Current velocities varied from 0.3 to 18.7 cm/s, averaging 6.1 cm/s [58]. The turbidity of the near-bottom layer was low.

In the area of Loki's Castle, the maximum suspended particulate matter signal was observed in the near-bottom layer up to ~250 m from the seafloor. That turbidity layer does not result solely from hydrothermal activity but instead reflects a combination of hydrothermal particles and resuspended sediments due to relatively high current velocities [32]. The area surrounding the Loki's Castle vent field is one of the few places where substantial amounts of sediment accumulate in the mid-ocean rift valley [48].

### 5.2. Specificity of Hosted Volcanogenic Deposits

Both vent sites are affected by neovolcanic activity [32]. Petrographic data show that samples of basalt nature occur in both hydrothermal systems, but there is a greater abundance of volcanic glasses with typical conchoidal glass fracture, hyaloclastites, and basalts in the Jan Mayen vent field area than in the Loki's Castle vent site. The young volcanites found in the Jan Mayen vent field area between hydrothermal sites are highly porous, which is typical of shallow volcanic outpourings [22]. The volcanic basement is covered by sandy sediments, indicating the relatively old age of the volcanic structures. The vast majority of basaltic glasses are subject to seafloor alteration, including palagonization [38,55]. The volcanic glasses have a vesicular structure with a porphyry and microporphyry matrix dominated by split plagioclase phenocrysts and idiomorphic clinopyroxene and olivine phenocrysts. The volcanic glasses have various alteration degrees in the Jan Mayen vent field area, from colorless, transparent, and dense to light yellow and fissured, with areas of palagonite and smectite, to a black isotropic opaque matrix composed entirely of smectite microfibers. According to the total alkalis classification [70], both whole-rock data and EDS data showed that the examined glasses belong to the tachylitic and tholeiitic basalt fields (see Supplementary Material, Table S2).

The dark brown, coarse volcanic glasses with minor alteration were occasionally found on the seafloor in the vicinity of Loki's Castle. The glassy fragments have a sparsely phyric texture, where a few microphenocrysts of plagioclase, olivine, and occasional pyroxene are identified. In altered areas and along fractures, there is a fibrous ochre material, palagonite. Our data showed that the examined glasses belong to the tholeiitic basalt field. In terms of petrochemical characteristics, the magmatism of the Mohns Ridge differs little from similar structures of the MAR type between latitudes of 34° and 48° N [71], i.e., the area influenced by the Azores hot spot.

### 5.3. Controls on Barite Mineralization in Hydrothermally Altered Sediments

In modern hydrothermal systems occurring in hotspots [22,24,32,72], barite is often represented in association with high-temperature minerals, which is also to be expected for the white smokers of the Jan Mayen vent fields. Barite has a lesser extent for the black smokers of the Loki's Castle vent field. The three studied vent fields are mafic-hosted systems. Basaltic rocks of the Jan Mayen vent fields have a characteristic E-MORB signature and are therefore enriched in barite, in contrast to the Loki's Castle vent field, which has a characteristic D-MORB signature [38], where barite is an accompanying mineral. Occasionally, barite occurs as a gangue mineral in the pyrite → gypsum → barite → goethite → talc mineral association of the Loki's Castle sediments. In the Troll Wall, barite is identified as a gangue mineral in the chalcopyrite → pyrite → sphalerite → barite → marcasite mineral association. The occurrence of a second generation of chalcopyrite, sphalerite, and barite and a third generation of pyrite in the association that overprints previous generations suggests temporal fluctuations in the temperature of mineralizing fluids [73]. The crystalline precipitation of finely crystalline sphalerite on large barite crystals suggests a possible increase in the temperature of hydrothermal fluid (Figure 7e). In the Perle and Bruse vent field, barite is one of the gangue minerals in the pyrite → gypsum → barite mineral association.

Thus, nonmetallic minerals predominate in the vent fields studied, and sulfide minerals are most often represented as disseminated fine grains and/or colloform textures. Complex intergrowths, replacements, and recrystallizations of the minerals are revealed as resulting from a relatively dynamic and locally chaotic environment of sulfide precipitation in altered sediments [37,74]. The Jan Mayen and Loki's Castle vent fields are pyrite-dominated among sulfide phases, accompanied by marcasite, sphalerite, and chalcopyrite. Other sulfide minerals are much less abundant; wurtzite is only found in Troll Wall; galena, a Pb-bearing mineral phase, is detected in both sites; and famatinite, a Cu-Sb-bearing mineral, is found in Loki's Castle. The selenium enrichment in the sediments of Loki's Castle is obviously caused by its concentration in galena, sphalerite, and pyrite, i.e., relatively high-temperature sulfide minerals. The arsenic enrichment in the sediments of the Perle and Bruse is caused by its co-precipitation with the hydrous ferric oxide fraction of the particles, i.e., the lower-temperature mineral phase of Fe-oxides [75].

A basic explanation of the mineralogical and geochemical diversity of deposits in the studied systems is the degree of maturity of the ore-generating hydrothermal systems, which is determined by the stages of mineral depletion of source rocks, beginning with the depletion of olivines, pyroxenes, and other dark-colored minerals and finishing with feldspars [76]. The formation of the opal, barite, Fe- and Zn-rich edifices of the Jan Mayen and the anhydrite, talc, Fe- and Cu-rich edifices of Loki's Castle is confined to this one of the last stages of depletion. According to [38], Loki's Castle may be a more mature vent field than Jan Mayen. The mineralogy of pyrrhotite-rich surface sediments of the Loki's Castle vent field suggests the first stage of alteration of seafloor sulfide sediments, halmyrolysis, according to [76], where two stages of hydrothermal sulfide halmyrolysis have been identified. Pyrite- and marcasite-rich surface sediments of the Troll Wall vent field are attributed to the transition between the first and second stages of halmyrolysis. Finally, the mineralogy of the surface sediments of the Perle and Bruse vent field exposed to the precipitation of Mn oxyhydroxides and barite suggests the second stage of halmyrolysis related to the oxidation of Fe(II). Thus, the three studied hydrothermal settings differ from each other not only in PT conditions, maturity, and alteration under oxidized conditions but also demonstrate the geodiversity of hydrothermal processes within each system.

The barite under study has the typical morphology characteristic of hydrothermal barite [1,5] and the same shape of crystals. Barite is a good indicator of proximal hydrothermal activity. It is a typical hydrothermal product, as suggested by the high Ba content in mineralizing fluid and the high $SO_4^{2-}$ content in seawater. The predominant sulfate phase in the sediments is jarosite → gypsum → barite. Barite is the predominant sulfate phase and one of the dominant minerals of hydrothermal origin in both studied Jan Mayen vent

fields. This is consistent with the Ba-rich hydrothermal fluids [38] penetrating the E-MORB host rocks. Barite is more common in the Jan Mayen sediments and has larger crystals than in Loki's Castle. The predominant sulfate phase in the Loki's Castle sediments is the same (i.e., jarosite → gypsum → barite), but in the dominant talc hydrothermally derived substrate of the sediments. According to published data, e.g., [32,37,38], the substrate of chimney edifices dominated by talc and anhydrite includes barite and amorphous silica caps. Consequently, in the Loki's Castle deposits, barite is a subordinate mineral, but its content can reach the high values corresponding to a major mineral, with the occasional occurrence of barite-rich mineral assemblages. That is consistent with the Ba-depleted hydrothermal fluids, which can be attributed to the D-MORB host rocks [38] and other PT conditions in contrast to the Jan Mayen vent fields. However, the results of thermodynamic modeling by Melekestseva et al. [25] show that the interaction of seawater with Ba-enriched basalt is not sufficient to produce a Ba-rich fluid and significant barite deposition on the seafloor. The evidence for fluid phase separation before barite precipitation makes this constraint even stronger. Nonetheless, the study of FIs provides additional insight into the precipitation conditions of barite-rich massive sulfides in the hydrothermal systems of the AMOR.

*5.4. Features of Hydrothermal Systems on the Mohns Ridge: Evidence from Fluid Inclusions Hosted in Barites*

Our data contribute to the few studies investigating FIs hosted in anhydrite by Sahlström et al. [37], where data are available, and in amorphous silica and barite by Strmic Palinkas et al. [39], where data are lacking in the hydrothermal systems on the Mohns Ridge. Here, the investigated FIs in barite were two-phase and represented a liquid-salt solution with a low-density vapor bubble. No significant differences were found between FIs hosted by barite crystals from the studied vent fields.

For a more correct comparison of the fluid temperatures of the studied hydrothermal fields, a box-whisker plot was used, which was previously used to compare the physical and chemical parameters of orogenic gold deposits of different ages [28]. The plot (Figure 10) shows that the higher temperature of the fluid corresponds to the Loki's Castle vent field, followed by the Perle and Bruse vent field, and finally the Troll Wall vent fields. No correlation between salinity and pressure-corrected temperature of FIs has been revealed, as, for example, such a correlation was found for FIs in other minerals in other vent fields of the MAR [77]. A general cooling trend with dilution of the Fåvne hydrothermal fluid down to 4.2 wt.% NaCl eq. was previously noticed [37] as well. The absence of a temperature trend associated with changes in salinity may indicate the presence of a single main fluid source (seawater) and a lack of participation of a noticeable share of hydrothermal ore-bearing fluids in the barite mineralization at different temperatures, and it is a characteristic of volcanogenic massive sulfide deposits [27]. Earlier, it was already noticed [39] that FIs entrapped in barite from Loki's Castle show compositions close to seawater, in contrast to amorphous silica, which hosts two-phase FI assemblages that preserve the characteristics of hydrothermal ore-bearing fluids.

5.4.1. Troll Wall Vent Field

The FIs study showed that fluids with a salinity ranging from 2.8 to 4.4 wt.% NaCl eq. circulated in the mineral-forming system in the Troll Wall vent field, and the pressure-corrected trapping temperatures were 119–276 °C. The direct measurements [32,33] showed that the maximum temperature of the fluids discharged onto the seafloor at the Troll Wall vent field was 270 °C. This value is close to the maximum temperature obtained by FI's study. Moreover, FI's research has shown that mineral formation at lower temperatures (119–260 °C) also occurs in hydrothermally altered vent-field sediments. The salt content of the mid-temperature fluids emanating from the Troll Wall differs from seawater; the measured salinity ranges from 82% to 138% of seawater. However, the mean value of salinity is slightly higher than seawater at 3.5 ± 0.36 wt.% NaCl eq. (here and after ±0.36

is the average deviation of the data from the mean value), and the median value (50% quartile) is 3.4 wt.% NaCl eq. It is known [38] that the vent fluids that are discharged from the relatively shallow Jan Mayen vent fields are undergoing phase separation (boiling) at the seafloor. At this phase, the fluid undergoes additional modification and is partitioned into a low-salinity, vapor-rich phase and a metal-rich brine phase [10,23,78].

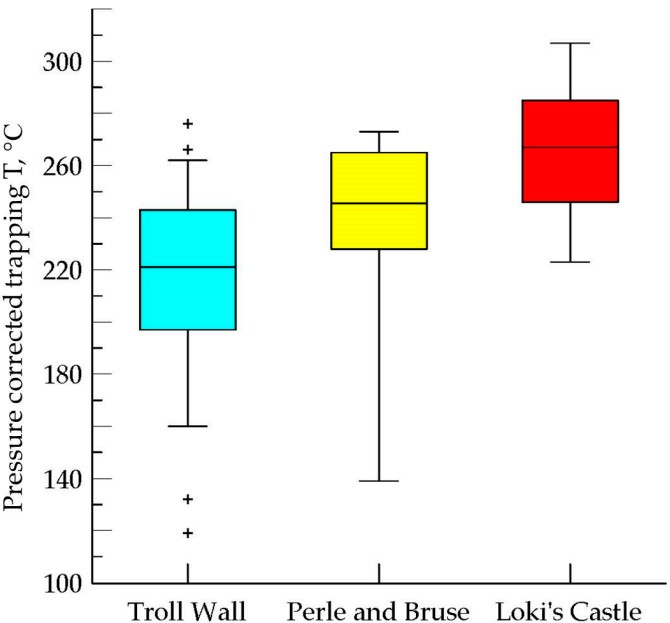

**Figure 10.** Comparison of trapping temperatures (pressure corrected) for fluid inclusions in barite disseminated in hydrothermally altered sediments of the studied vent fields. The whiskers correspond to the 5th and 95th percentiles. The presence of potential outliers, that is, points that fall beyond the 95th and 5th percentile limits, is indicated by the symbol "+" in the plot.

Thus, several immiscible phases and fluids of different compositions were circulating in the mineral-forming system of the Troll Wall vent field, where barite and associated Fe and Zn sulfides and Fe-Si oxyhydroxides were deposited. It is known that the homogenization temperature of FIs can differ from the crystallization temperature of the host mineral by some value, which depends primarily on the pressure [36]. The difference between the maximum crystallization (pressure-corrected trapping) temperature and the maximum homogenization temperature is 30 °C. That corresponds to a pressure of 430 bar (approximately 4300 m below the seafloor). Possibly, we may estimate the depth of fluid–rock interaction at about 3700 m below the seafloor.

### 5.4.2. Perle and Bruse Vent Field

The FI studies showed that fluids with a salinity ranging from 3.1 to 4.2 wt.% NaCl eq. circulated in the mineral-forming system in the Perle and Bruse vent field, and the pressure-corrected temperatures of mineral formation are 139–273 °C. The direct measurements [56,79] of emitted fluids revealed temperatures between 229 and 270 °C and a lower chloride content than in seawater, indicating that the fluids had undergone phase separation. The present-day venting fluids at the Perle and Bruse vent field are less saline (about 2.6 wt.% NaCl eq.) than our data derived from FIs. Nonetheless, the salt contents derived from FIs differ from seawater; the measured salinity ranges from 97% to 132% of seawater. The mean and median values of fluid salinity (3.7 ± 0.26 wt.% NaCl eq. and 3.8 wt.% NaCl eq., respectively) are slightly higher than seawater. Moreover, the low-temperature venting is evidenced by the presence of birnessite at the Perle and Bruse vent field, ~2 km east of the Troll Wall vent field [21,53]. It is noteworthy that the Perle and Bruse vent field is located not in the rift valley but on the flank of the eastern AVR massif, at the base of the fault scarp.

### 5.4.3. Loki's Castle Vent Field

The FI studies showed that fluids with a salinity ranging from 2.8 to 4.4 wt.% NaCl eq. circulated in the mineral-forming system of the Loki's Castle vent field, and the pressure-corrected temperatures of mineral formation are 223–307 °C. The salt content of the fluids emanating from Loki's Castle differs from seawater; the measured salinity ranges from 88% to 138% of seawater. However, the mean and median values of salinity are close to seawater (3.5 ± 0.37 wt.% NaCl eq.). Previously [33], according to direct measurements from the chimney edifices, it had been reported that the fluid salinity was less than that of seawater. Chloride concentrations different from seawater indicate that the venting fluid has undergone phase separation processes. According to [78], the Loki's Castle fluids did not phase separate at the seafloor like in the Jan Mayen vents, but phase separation occurred in or above the high-temperature reaction zone, where low-density vapors rise in the hydrothermal cell and high-density brines either accumulate at the base of the circulating system or rise slowly. Baumberger et al. [33] argued that the minimum depth and temperature to produce a vapor phase with about 2.9 wt.% NaCl eq. are 650 m below the seafloor [33].

### 5.4.4. Comparative Study of Fluid Inclusions Hosted in Hydrothermal Barites from the Mohns Ridge with Similar Products from Other Hydrothermal Systems

A wide variation in crystallization temperature is characteristic of mineral formation in the hydrothermal vent fields on the Mohns Ridge. Our estimates of pressure-corrected temperatures are consistent with the present-day temperature at the studied vents but represent the wider temperature fluctuations of mainly moderately-temperature venting of (Fe, Ba)-bearing hydrothermal fluids. The study of FIs shows that products of barite formation at lower temperatures are also found in hydrothermally altered sediments of these vents, i.e., minerals continued to crystallize under fluid temperatures falling to 119 °C and 139 °C for the Troll Wall and Perle and Bruse, respectively. The lowest temperature values may reflect the deposition of late-stage barite. The lowest temperature values obtained from the FIs are approximately 50–150 °C lower than the present-day direct temperature measurements. Significant changes in the temperature of mineralizing fluids have previously been established for a wide range of hydrothermal systems [77]. Moreover, measurements at one minute intervals with an autonomous thermometer at a black smoker of the Rainbow vent field (low-spreading MAR) showed that the temperature repeatedly fluctuated between 250 °C and 348 °C during 35 min [80]. The relatively wide pressure-corrected temperature range obviously indicates the mixing of seawater and hydrothermal fluids in the highly porous massive sulfides and basalts, e.g., [81]. The fluid–rock interaction at ~2 km below the seafloor was suggested for Loki's Castle [32] and possibly at ~3.7 km for the Jan Mayen vents [74].

Two-phase fluids are subject to significant temperature fluctuations, which can lead to significant changes in fluid composition, primarily in salinity. The salinity of the fluids hosted in barites in the vent fields studied ranges from 2.6 to 4.4 wt.% NaCl eq., which is only relatively close to the salinity of the ambient seawater. A box-whisker plot was used to demonstrate the salinity variations of fluids in the studied hydrothermal systems (Figure 11). In the plot, the median value of salinity varied from 3.4 to 3.8 wt.% NaCl eq., with the highest median value corresponding to fluids circulating in the Perle and Bruse, which is slightly higher than direct measurements of the present-day venting Bruse fluids [56,79]. There is no trend between cooling the hydrothermal fluid and diluting it down to 2.6 wt.% NaCl eq., and therefore a fluid with low salinity (compared to ambient seawater) cannot simply be the result of mixing with cold seawater.

Thus, the salt contents of hot fluids venting from the Mohns Ridge hydrothermal systems generally differ from seawater, with measured salinities ranging from 82% to 138% of seawater and showing much fewer variations than those reported for the mid-ocean ridges, ranging from 10% to 200% of seawater [82]. Analysis of FIs trapped in anhydrites in the Fåvne vent field at ~3000 m depth in the central part of the Mohns Ridge [37]

showed that fluid salinities were typically about twice the salinity of the ambient seawater and ranged from 4.2 to 8.0 wt.% NaCl eq. The reported salinity of the trapped Fåvne fluid is higher than our data derived from FIs in barites. The low-salinity fluids down to 2.6–3.1 wt.% NaCl eq. trapped in barites of studied vents could thus represent condensed vapors generated by phase separation (boiling) of a higher-salinity aqueous fluid below the seafloor [23,25]. This process produces a heavy saline brine, which mostly remains at depth, and a low-salinity vapor phase, which migrates toward the seafloor [33,78,83]. Low-salinity and low-density vapors are ascended buoyantly in the hydrothermal system, mixed with cold seawater, and finally emanate at the seafloor, producing low-salinity (compared to seawater) vents [78]. The wide variations in fluid temperature and salinity in the studied vent fields can be explained by both temporal fluctuations, which have been verified by [33], and the circulation of several immiscible phases and fluids of different compositions in hydrothermal systems. The latter hypothesis is supported by published data on the presence of areas with diffuse venting of low-temperature fluids (approximately 2–54 °C) in the vicinity of chimneys with focused venting [13,34,55,56]. The diffuse venting emanating from sediments is obviously derived from the relatively high-temperature endmember fluids. Evidently, that is a common phenomenon for hydrothermal systems in the AMOR and MAR [32,40].

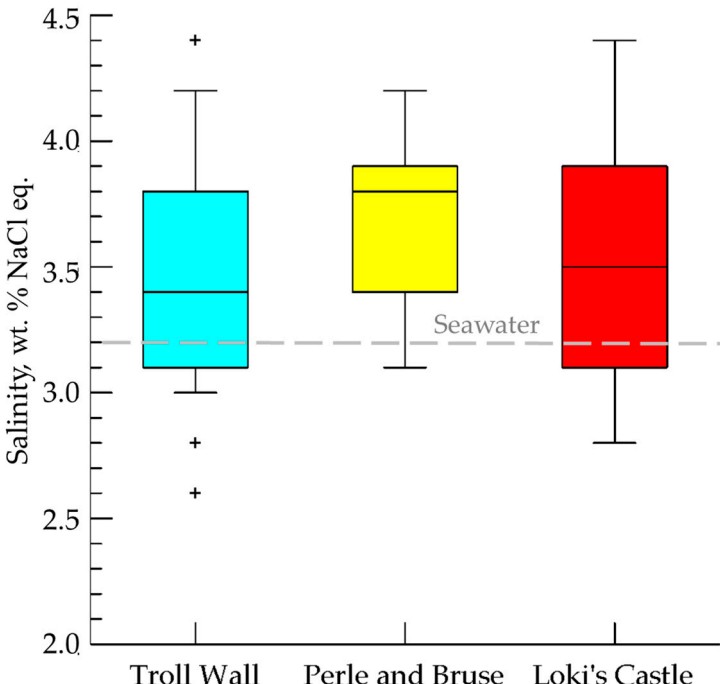

**Figure 11.** Comparison of salinity for fluid inclusions in barite disseminated in hydrothermally altered sediments of the studied vent fields. A gray dotted line indicates the salinity of ambient seawater. The whiskers correspond to the 5th and 95th percentiles. The presence of potential outliers, that is, points that fall beyond the 95th and 5th percentile limits, is indicated by the symbol "+" in the plot.

Thus, two processes play a major role in fluid compositions: phase separation and the composition of the basement rocks [40]. FIs with high (up to 50 wt.% NaCl eq.) salinity are believed to be of magmatic origin, while FIs with low and moderate salinity (<1.0 and up to 15–20 wt.% NaCl eq.) are attributed to phase separation followed by the segregation of the vapor and brine phases [78,84,85]. A comparative study with similar products from other hydrothermal vent fields is presented in Table 4 in an attempt to understand how the systems studied relate to the overall context. The salinity of the fluids differs from that of the ambient seawater, which is assumed to be the major component of the mineralizing fluid in generally accepted models of submarine hydrothermal systems [77].

**Table 4.** Temperature and salinity of primary hydrothermal fluids at the Mid-Atlantic Ridge (MAR) and Mohns Ridge segment of the Arctic Mid-Ocean Ridge (AMOR) according to published and own data.

| Vent Field | Location | Depth, m | Deposit Type | Fluid Salinity, wt.% NaCl eq. | Fluid T *, °C | Comment | Reference |
|---|---|---|---|---|---|---|---|
| Troll Wall | 71°18′ N, Mohns Ridge, AMOR | 550 | Chimney edifices | n.d. ** | 260–270 | Direct measurements. | [32] |
| | | 512–600 | Hydrothermally altered sediments | 2.6–4.4 | 119–276 | Primary and secondary FIs in barite. Fluid phase separation (FPS) is indicated. | This work |
| Bruse | 71°18′ N, Mohns Ridge, AMOR | 561 | Barite-rich chimney edifices | ~2.6 | 240–242 | Direct measurements: maximum temperature. FPS is indicated. | [56] |
| | | 560 | Barite-rich chimney edifices | ~2.5 | 229–270 | Direct measurements. FPS is indicated. | [79] |
| Perle and Bruse | 71°18′ N, Mohns Ridge, AMOR | 620 | Hydrothermally altered sediments | 3.1–4.2 | 139–273 | Primary and primary–secondary FIs in barite. FPS is indicated. | This work |
| Soria Moria | 71°15′ N, Mohns Ridge, AMOR | ~700 | Chimney edifices | ~3.1 | 50–270 | Direct measurements. | [79] |
| Loki's Castle | 73°30′ N, Mohns Ridge, AMOR | ~2000 | Talc-anhydrite chimney edifices | ~2.9 | 280–317 | Direct measurements. FPS is indicated. | [33] |
| | | 2376 | Hydrothermally altered sediments | 2.8–4.4 | 223–307 | Primary and secondary FIs in barite. FPS is indicated. | This work |
| Fåvne | 72°45′ N, Mohns Ridge, AMOR | ~3000 | Seafloor massive sulfide | 4.2–8.0 | 200–291 | Primary FIs in anhydrite. Fluid did not experience significant mixing with seawater. | [37] |
| Menez Gwen | 37°50′ N, MAR, influenced by the Azores hot spot | 847–871 | Anhydrite and Anhydrite-barite edifices with disseminated sulfides | 2.2–2.3 | 271–284 | Direct measurements. FPS is indicated. | [22] |
| Lucky Strike | 37° N, MAR, influenced by the Azores hot spot | 1618–1730 | Sulfate-sulfide deposits | 2.9–3.5 | 202–325 | Direct measurements. Evidence for FPS. | [72] |

**Table 4.** *Cont.*

| Vent Field | Location | Depth, m | Deposit Type | Fluid Salinity, wt.% NaCl eq. | Fluid T *, °C | Comment | Reference |
|---|---|---|---|---|---|---|---|
| Rainbow | 36°14′ N and 33°54′ W, MAR | 2270–2320 | Sulfide edifices and fields associated with serpentinites | 4.5–7.7 | 295–370 | Primary FIs in anhydrite are associated with marcasite and chalcopyrite. FPS is proposed. | [80] |
| Broken Spur | 29°10′ N, 43°10′ W, MAR | ~3000 | Sulfide chimney, walls of a tube | 3.0–6.3 | 259–406 | Primary FIs in anhydrite. FPS is indicated. | [77] |
| TAG | 26°08′ N, MAR | 3670 | Volcanogenic massive sulfides, edifices | 1.2–5.1 | 187–390 | Primary FIs in anhydrite. FPS is indicated. | [86] |
| | | | Breccia, veins, quartz, massive granular pyrite | 1.9–6.2 | 212–390 | Primary FIs in anhydrite and quartz. FPS is indicated. | [87] |
| Logachev-1 | 14°45′ N, MAR | 2970 | Sulfide chimney, walls of a tube | 4.2–16.2 (max 26) | 271–365 | Primary FIs in anhydrite. FPS is indicated. | [77] |
| | | | | 1.9–6.2 | 212–390 | Primary FIs in anhydrite and quartz. FPS is indicated. | [87] |
| Semenov-1 | 13°30′ N, MAR | 2400–2950 | Barite-rich massive sulfides | 0.6–3.8 | 83–244 | Primary FIs in barite. FPS is indicated. | [25] |
| Ashadze 1, Long Chimney | 12°58′ N, MAR | 4080 | Sulfide edifices | 5.0–7.0 | 295–345 | Primary FIs in anhydrite. FPS is indicated. | [88] |
| | | | Embryo sulfide edifices | 5.0–7.8 | 235–355 | Primary FIs in anhydrite. FPS is indicated. | [88] |
| JADE | Central Okinawa Trough, NW Pacific. Back-arc basin. | 1300–1600 | Sulfide-sulfate chimneys and mounds. Stockwork mineralizations. | 4.4–9.6 | 220–320 | Primary FIs in gypsum and barite. | [89] |
| | | | | 2.0–15.0 | 270–360 | Primary FIs in sphalerite. FPS is indicated. | [11] |
| Endeavour Segment | Juan de Fuca Ridge, NE Pacific | 2050–2700 | Barite-rich sulfide edifices | 5.7–9.4 | 124–283 | Primary FIs in barite. FPS is not confirmed. | [18] |

**Table 4.** *Cont.*

| Vent Field | Location | Depth, m | Deposit Type | Fluid Salinity, wt.% NaCl eq. | Fluid T *, °C | Comment | Reference |
|---|---|---|---|---|---|---|---|
| Vienna Wood | 3°10′ S, 150°17′ E, Manus basin, SW Pacific | ~2500 | Barite-silica-sulfide chimney, top of an active tube | 4.7–7.6 | 165–235 | Primary and secondary FIs in barite. FPS is indicated. | [77] |
| | | | | 5.3–7.2 | 242–324 | Primary FIs in anhydrite. FPS is indicated. | [77] |
| | | | | 4.1–8.5 | 160–247 | Primary FIs in anhydrite are associated with chalcopyrite and sphalerite. FPS is proposed. | [80] |
| Franklin seamount | 9°55′ S, 151°50′ E, Woodlark basin, SW Pacific | 2143–2366 | Barite-sulfide edifices | 2.7–6.9 | 203–316 | Primary FIs in barite. FPS is indicated. | [77] |
| Lau back arc basin | 19°20′–22°50′ S, Valu Fa Ridge, Tonga arc, SW Pasific | 1735 | White Church, barite-sulfide chimneys | 1.6–2.6 | 170–230 | Primary FIs in barite. FPS is indicated. | [23] |
| Hook Ridge | 62°11′ S, 57°15′ W, Bransfield Strait, Antarctica | 990 | Chimney and massive barite slab. Arc-shaped composite volcano | 0.9–4.2 | 132–310 | Primary FIs in barite. FPS is indicated. | [81] |

* Temperatures from FIs reflect pressure-corrected homogenization temperatures, unless noted otherwise. ** n.d.—not determined.

Venting fluids are likely to be ultimately remixed fluids that are less saline than the initially separated brine fluid, as observed in most hydrothermal systems [23,25,83,84]. Most hydrothermal fluids have salinity variations of 2–6 times, but typically salinity is at least 5–6 times lower than that of ambient seawater and twice as much as that of ambient seawater. In the model by Bischoff and Rosenbauer [90], mid-ocean ridge hydrothermal systems are composed of a single-pass seawater cell overlying a dense, high-temperature brine cell with salinities that are at least five times seawater. The high salinity fluids are more evident from intermediate and fast-spreading rides due to the assimilation of Cl-rich crust during crystallization in a shallow magma chamber [91]. It is believed that volcanic processes in low-spreading ridges developed above small local magmatic chambers (volcanic phase) give way to a long-lasting tectonic phase when the magmatic chamber disappears [80].

The FIs in the barites of the hydrothermal systems under study show a salinity not exceeding 1.4 times the salinity of ambient seawater. At the deeper Fåvne vent field [37], the salinity exceeds that of the ambient seawater by a factor of 2.3. The lowest calculated salinity of FIs in the Troll Wall is approximately 3–4 times higher than the lowest salinity of barite FIs in the Hook Ridge, Bransfield Strait (0.9 wt.% NaCl eq. [81]), and Semenov-1 (0.6 wt.% NaCl eq. [25]), respectively, which is the lowest reported value for seafloor hydrothermal fields. Our FIs data derived from barites and published data derived from anhydrites [37] in the discussed hydrothermal vent fields on the Mohns Ridge are consistent with 90% of available data on homogenization temperature–salinity pairs from volcanogenic massive sulfide ore deposits summarized by Bodnar et al. [27] (Figure 12). Therefore, low- and moderate-salinity fluids circulate in the hydrothermal systems on the Mohns Ridge, but according to the FI studies in barites, fluids of moderate salinity, relatively close to the salinity of the ambient seawater, predominate in these systems. Despite the temporal fluctuations in temperature and composition of mineralizing fluids revealed by Baumberger et al. [33] and proposed in our study, the hydrothermal systems on the Mohns Ridge represent a more stable chemical environment compared to those at fast-spreading ridges [92]. This constant emanation of hydrothermal fluids makes these hydrothermal vents a relatively stable source of Ba, Fe, Zn, Cu, and other trace elements for the Arctic Ocean and the subpolar North Atlantic.

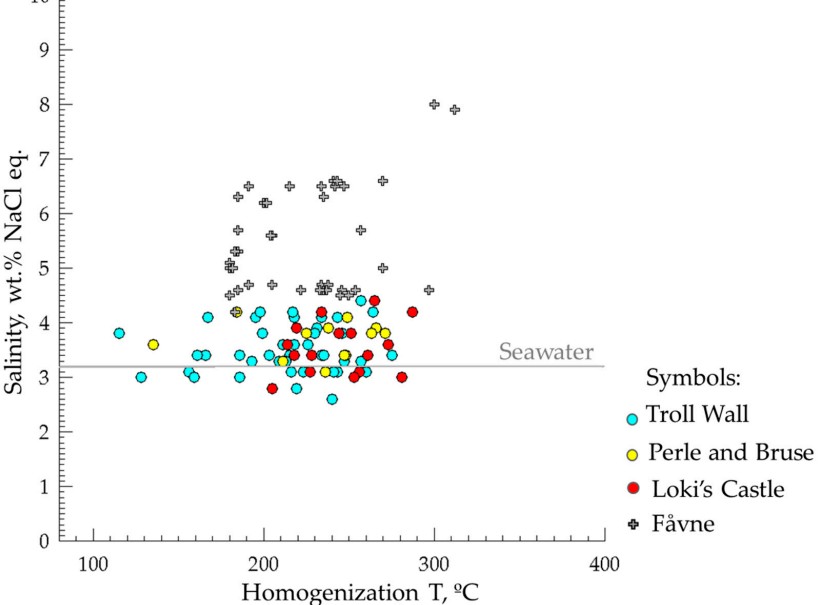

**Figure 12.** Trapping temperatures (pressure corrected) and corresponding salinities for fluid inclusions in barite (Troll Wall, Perle and Bruse, and Loki's Castle) and anhydrite (Fåvne) from volcanogenic massive sulfide deposits of the Mohns Ridge. The horizontal gray line indicates the salinity of ambient seawater. Data on fluid inclusions in anhydrite from the Fåvne vent field are taken from [37].

## 6. Conclusions and Perspectives

The study of fluid inclusions hosted in barites from hydrothermally altered sediments of the Mohns Ridge represents typical data for the general fluid characteristics of the volcanogenic massive sulfide deposits. Our findings contribute to the few studies [37,39] investigating FIs hosted in anhydrite, amorphous silica, and barite in the hydrothermal edifices on the Mohns Ridge. The FIs are mostly two-phase, liquid–plus–vapor that homogenize to the liquid phase, with salinities less than 8 wt.% NaCl eq. The FIs in barites from the Jan Mayen and Loki's Castle hydrothermal sediments have salinities of less than 4.4 wt.% NaCl eq. and show a predominance of Fe, Mg, and Na chlorides in the fluids. The salinity of FIs is typically in the range of 2.6–4.4 wt.% NaCl eq., or twice greater (Fåvne vent field) than or 1.2 times less than the salinity of the ambient seawater. Thus, the studied hydrothermal systems are characterized by low- and moderate-salinity fluids and undergo fluid phase separation (boiling) below the seafloor according to generally accepted models. The phase separation produces a lower salinity phase, represented in ~24% of the FIs studied. The relatively narrow range of salinity variations measured for FIs indicates small variations in the chemistry of the hydrothermal fluids due to boiling. The revealed values of fluid salinity correspond to ultraslow-spreading ridges. The hydrothermal systems on the Mohns Ridge represent a more stable chemical environment compared to those at fast-spreading ridges.

Barite precipitates from moderate temperature (from 119 °C to 307 °C) sulfide–sulfate–aqueous fluids. Pressure-corrected temperatures of mineralizing hydrothermal fluids vary over a wider range than was previously suggested on the basis of direct measurements. The lowest temperature values obtained from FIs in barite are ~50–150 °C below the present-day temperature measured using a remotely operated underwater vehicle during cruises with the G.O. Sars research vessel. The variations in mineralizing fluid temperature and salinity in the studied hydrothermal systems may reflect: (i) the deposition of late-stage barite; (ii) temporal fluctuations of ore-bearing fluids; and (iii) the circulation of several immiscible phases and fluids of slightly different compositions. The absence of a temperature trend related to changes in salinity is a characteristic feature of volcanogenic massive sulfide deposits.

The data obtained emphasize the relatively stable characteristics of fluids circulating in the studied hydrothermal systems in general and the slightly higher salinity of fluid in the Perle and Bruse vent field in contrast to present-day fluid measurements. Consequently, this study has extended the dataset for fluid characterization and improved our understanding of the hydrothermal systems on the Mohns Ridge. Therefore, FI studies hosted in the minerals of modern seafloor hydrothermal sites are a reliable approach not only to fill a gap in the direct measurements of hydrothermal fluids but also to provide more detailed knowledge of the mineralizing fluids circulating in the hydrothermal system and the nature of their evolution.

**Supplementary Materials:** The following supporting information can be downloaded at: https://www.mdpi.com/article/10.3390/min13091117/s1, Table S1. EDS data of volcanic glasses collected from hydrothermally altered sediments of the Jan Mayen vent field area; Table S2. XRF major element composition and Ba (wt.%) of the surface sediment samples and hyaloclastites.

**Author Contributions:** Conceptualization, M.D.K. and A.Y.L.; methodology, V.Y.P. and M.D.K.; software, V.Y.P.; validation, V.Y.P., M.D.K. and A.Y.L.; formal analysis, V.Y.P., M.D.K., O.M.D., A.A.K., K.S.I. and V.Y.K.; investigation, M.D.K., A.A.K. and B.V.B.; resources, M.D.K. and V.Y.P.; data curation, M.D.K.; writing—original draft preparation, M.D.K.; writing—review and editing, M.D.K., V.Y.P., B.V.B. and A.Y.L.; visualization, M.D.K. and B.V.B.; supervision, M.D.K.; project administration, M.D.K.; funding acquisition, M.D.K. All authors have read and agreed to the published version of the manuscript.

**Funding:** This research was funded by the Russian Science Foundation, grant number 20-17-00157, https://rscf.ru/project/20-17-00157/. Field studies were carried out within the state assignment of the Shirshov Institute of Oceanology, Russian Academy of Sciences, subject number FMWE-2021-0006.

**Data Availability Statement:** The original contributions presented in the study are included in the article and Supplementary Material; further inquiries can be directed to the corresponding author.

**Acknowledgments:** We acknowledge the crew and participants of the 75th cruise of the RV Akademik Mstislav Keldysh carried out in June 2019 for helping acquire the sediment samples. We are also grateful to Georgii Malafeev and Sergey Isachenko (IO RAS) for assisting with collecting the samples. We thank Andrey Boev and Alexander Filippov (IO RAS) for their assistance in the analysis process.

**Conflicts of Interest:** The authors declare no conflict of interest.

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
