# Peer review of "Fluid Inclusion Studies of Barite Disseminated in Hydrothermal Sediments of the Mohns Ridge"

_minerals, doi:10.3390/min13091117_

Round 1

Reviewer 1 Report

The paper presents a study of moderate interest based on a detailed presentation of the background of the problem and the experimental data.  The presentation tends to be wordy and sometimes repetitious especially in the conclusions.  A total of 94 references were cited, a lot for the topic, and some seem to be clearly excessive. There are problems with Tables 1 and 3, but serious issues concerning use of Kullerud's sphalerite geothermometer and conclusions about the stability of marcasite. I have discussed these in an attached file containing detailed comments.  I have made numerous comments directly to the text, which point out many typos and misspellings as well as other minor points.  

The authors' use of English displays good fluency with a few minor problems that I have noted on the text.

Author Response

Dear Reviever,

We would like to submit for publication in Minerals our revised manuscript entitled “Fluid inclusion studies of barite disseminated in hydrothermal sediments of the Mohns Ridge”. We sincerely appreciate Reviewer for their valuable comments and suggestions, which helped us in improving the quality of the manuscript. So, we have made a substantial revision of the manuscript. We corrected the text and Tables 1, 2 and 3. All corrections are highlighted by blue color. We revised Figures 1, 2, 8, and 11.  We appreciate the time and effort you spent on reading our manuscript.

On behalf of the co-authors,

Marina Kravchishina

Reviewer 2 Report

The manuscript titled “Fluid Inclusion Studies of Barite Disseminated in Hydrothermal Sediments of the Mohns Ridge” is interesting and well prepared, which provide a plenty of significant fluid inclusion studies of barite and mineralogy and petrology of host rocks. The manuscript can be published after a minor revision. Here are some comments for authors to consider.

(1)   Introduction: You should raise several questions here which may be discussed in the last section of the text (Discussion). The sub-titles in your discussion are not clearly, which seems to be too general, unlike to answer the raised questions.

(2)   Fig.1: The vertical coordinates (Y-axis) has the depths of negative values, not positive values.

(3)   Table 2: Please list the range of analyzed values for homogenization temperatures, ice melting temperatures, .........

(4)   Discussion: Your sub-titles here are “5.1. Hydrothermal Vent Field Settings”, “5.2. Barite Hosted Rocks”, “5.3. Characterisation of Mineralised Material”, and “5.4. Fluid Inclusions Hosted in Barites”. The readers could not find what questions raised in "Introduction". I suggest you may raise a few questions in "Introduction", then answer these questions in this section.

(5)   Conclusions:Conclusions may be rewritten in brief. Some sentences seem to be explanations instead of conclusions, which are seen in the previous sections of the paper.

Can be improved. Some paragraphs are tedious. 

Author Response

Dear Reviewer,

We would like to submit for publication in Minerals our revised manuscript entitled “Fluid inclusion studies of barite disseminated in hydrothermal sediments of the Mohns Ridge”. We sincerely appreciate you for valuable comments and suggestions, which helped us in improving the quality of the manuscript. So, we have made a substantial revision of the manuscript. We corrected the text and Tables 1, 2 and 3. All corrections are highlighted by blue color. We revised Figures 1, 2, 8, and 11.  The poin by point resposes are in the attachment. 

On behalf of the co-authors,

Marina Kravchishina

Round 2

Reviewer 1 Report

The revisions appear to  adequately deal with all comments made on the original submission.  However, I still feel that some of the referencing is excessive and not always appropriate.
